# Principled Confidence Estimation for Deep Computed Tomography

**Matteo Gätzner** [* 1]    **Johannes Kirschner** [* 2]

## Abstract

We present a principled framework for confidence estimation in computed tomography (CT) reconstruction. Based on the sequential likelihood mixing framework (Kirschner et al., 2025), we establish confidence regions with theoretical coverage guarantees for deep-learning-based CT reconstructions. We consider a realistic forward model following the Beer-Lambert law, i.e., a log-linear forward model with Poisson noise, closely reflecting clinical and scientific imaging conditions. The framework is general and applies to both classical reconstruction algorithms and deep learning methods alike, including U-Nets, U-Net ensembles, and generative Diffusion models. Empirically, we demonstrate that deep reconstruction methods yield substantially tighter confidence regions than classical reconstructions, without sacrificing theoretical coverage guarantees. Our approach allows the detection of hallucinations in reconstructed images and provides interpretable visualizations of confidence regions. This establishes deep models not only as powerful estimators, but also as reliable tools for uncertainty-aware medical imaging.

## 1. Introduction

Computed Tomography (CT) is a ubiquitous imaging technique essential for critical decision-making across medicine, manufacturing, and chip metrology (Kak & Slaney, 2001; De Chiffre et al., 2014). By reconstructing a high-dimensional 3D volume or 2D slice from a series of noisy X-ray projection measurements, CT allows practitioners to peer inside objects and probe matter, facilitating medical diagnostics, and quality control and inspection in industrial settings.

Traditionally, tomographic reconstruction is performed using analytical methods such as Filtered Backprojection (FBP; Bracewell & Riddle, 1967) or iterative algorithms such as those based on expectation-maximization (Shepp & Vardi, 1982). More recently, deep-learning-based approaches have achieved state-of-the-art performance (Zhang et al., 2025; Liu et al., 2024; Lin et al., 2023), using architectures such as U-Nets (Ronneberger et al., 2015) and diffusion models (Ho et al., 2020) to recover high-fidelity images even from sparse or low-dose data (Kiss et al., 2025). However, adoption of these methods in clinical and industrial settings remains limited due to concerns regarding robustness, interpretability of learned models, and the lack of theoretical guarantees compared to traditional reconstruction techniques (Rudin, 2019; Gottschling et al., 2025). Moreover, both classical and modern data-driven paradigms typically provide a single *point prediction* without uncertainty certificates or confidence regions.

The lack of uncertainty quantification is particularly problematic in safety-critical settings. In medical imaging, noise artifacts can be misinterpreted as pathologies, and conversely, small lesions may be smoothed out. Furthermore, generative models are notoriously prone to *hallucinations* — plausible-looking structures that are not supported by the measurement data (Antun et al., 2020; Bhadra et al., 2021). Without a measure of confidence, operators do not have a mechanism to distinguish between a reliable structural detail and a generative artifact. Additionally, the absence of uncertainty quantification forces a conservative approach to the inherent trade-off between dose and image quality. Because lower X-ray exposure results in higher measurement noise, and consequently, lower-fidelity reconstructions, scanning protocols default to fixed, high-dose schedules. While ensuring quality, this approach exposes patients to unnecessary radiation risks or damages radiation-sensitive objects, as the measurement protocol lacks the feedback loop to stop acquisition once sufficient information is obtained.

To address these challenges, we present a framework for *principled confidence estimation* in CT reconstruction. Moving beyond heuristic uncertainty measures, we aim to construct confidence sets with theoretical coverage guarantees — regions in the image space that are guaranteed to contain the ground truth image with a user-specified probability (e.g., 95%) w.r.t. the randomness in the observation data.

---

[*]Equal contribution  [1]Department of Mathematics, ETH Zürich, Zürich, Switzerland [2]Swiss Data Science Center, ETH Zürich, Zürich, Switzerland. Correspondence to: Matteo Gätzner <matteo.gatzner@gmail.com>, Johannes Kirschner <johannes.kirschner@sdsc.ethz.ch>.

*Proceedings of the 43rd International Conference on Machine Learning*, Seoul, South Korea. PMLR 306, 2026. Copyright 2026 by the author(s).

In this work, we build on likelihood-based inference and confidence estimation (Robbins & Siegmund, 1970), that exploit martingale properties of the sequential likelihood ratio to construct data-driven confidence sequences as measurement data are acquired. More specifically, we adopt the *sequential likelihood mixing* framework of Kirschner et al. (2025). This framework is uniquely suited to the physical forward model and acquisition protocol of CT, where data is naturally acquired as a sequence of projection measurements with a finite exposure time. By modeling the non-linear forward operator and Poisson noise inherent to X-ray interactions, we derive confidence sequences with provable finite-time coverage guarantees that are valid at any stopping time. The geometry of the confidence set is determined by the negative log-likelihood of the observed data sequence, and a data-driven confidence coefficient that measures the sequential test error of a sequence of reconstructions. The construction applies to any reconstruction method and likelihood function, and, crucially, yields tighter confidence sets given more accurate predictions. This allows us to integrate powerful deep learning estimators, such as U-Nets (Ronneberger et al., 2015) and diffusion models (Ho et al., 2020), to shrink the size of the confidence region, providing tight, informative uncertainty estimates while maintaining theoretical coverage guarantees.

**Contributions.** In this work, we present the first application of sequential likelihood mixing to computed tomography, to obtain finite-sample, anytime confidence regions with provable coverage guarantees that account for the non-linear physical forward model and Poisson noise statistics. We demonstrate that modern deep learning architectures (e.g., U-Nets, Ensembles, Diffusion Models) seamlessly integrate into this framework while preserving the theoretical validity of the constructed confidence region. We show empirically that leveraging the predictive power of these models enables substantially tighter confidence sets compared to classical baselines such as FBP and maximum likelihood estimation. Based on the confidence set construction, we propose a computationally efficient method to detect model hallucinations, flagging reconstructions that are statistically inconsistent with the observed projection data. We further develop methods to project high-dimensional confidence sets into visually interpretable pixel-wise uncertainty maps, and show that the resulting intervals are tighter at the nominal coverage level than those produced by a range of established UQ baselines including bootstrap and MCMC. We make our code publicly available[1]. Our work represents a significant step towards practical uncertainty certificates for deep computed tomography reconstructions, with downstream applications such as early stopping and data-driven experimental design (Barba et al., 2024).

---

[1] github.com/SwissDataScienceCenter/uqct

## 1.1. Related Work

**Uncertainty Quantification in Tomography.** Classical uncertainty quantification in tomography relies on analytic noise propagation (Qi & Leahy, 2000), bootstrap resampling (Dahlbom, 2001; Pereyra & Tachella, 2024) or Bayesian inference (Hanson & Wecksung, 1983; Chen et al., 2009; Higdon et al., 1997). While Bayesian methods model the full posterior (Zhou et al., 2020; Pedersen et al., 2022; Lee et al., 2024a), computational costs often necessitate approximations like ensembling (Lakshminarayanan et al., 2017), Monte Carlo dropout (Vasconcelos et al., 2023), Laplace approximation (Antoran et al., 2023) and accelerated Markov chain Monte Carlo (MCMC) algorithms (Durmus et al., 2018; Pereyra et al., 2020), some of which have been developed specifically in the context of tomography applications. For log-concave posteriors, Pereyra (2017) constructs scalable Bayesian credible sets as MAP level sets, which are structurally similar to our confidence sets but targeting posterior probability rather than frequentist coverage. Most prior methods generally lack rigorous finite-sample frequentist guarantees (Gillmann et al., 2021). One exception is Hoppe et al. (2025), who derive pixel-wise confidence intervals for MR reconstruction for a sparse Lasso estimator. CT-specific Bayesian UQ has further addressed view-angle uncertainty (Riis et al., 2021), defect detection in a Gaussian/linear regime (Christensen et al., 2024), and goal-oriented UQ for inclusion boundaries (Afkham et al., 2023).

**Confidence Sequences and Sequential Inference.** Our work is rooted in the theory of confidence sequences; sequences of confidence sets that are valid at arbitrary stopping times (Darling & Robbins, 1967; Robbins & Siegmund, 1970; Robbins, 1970), going back to the seminal work on sequential analysis (Wald, 1945). Unlike fixed-sample hypothesis tests or standard conformal prediction, confidence sequences allow for continuous monitoring of the data without inflating the error rate, a property known as *anytime validity*. More recent works build upon these ideas, in the context of universal inference and testing (Wasserman et al., 2020), kernel regression (Flynn & Reeb, 2024) and sequential decision-making (Lee et al., 2024c). Ramdas et al. (2023) provide a unified modern treatment of the game-theoretic framework underlying anytime-valid inference via e-values and test martingales. Clerico et al. (2025) study the application of regret bounds from online learning to establish tight statistical confidence bounds for exponential families. While likelihood-based confidence sets have been developed for various statistical settings, their application to high-dimensional inverse problems with deep learning priors appears to be novel. We specifically leverage the *Sequential Likelihood Mixing* framework proposed by Kirschner et al. (2025), that extends and unifies likelihood-based confidence estimation, and establishes con-

nections to Bayesian inference, variational inference, maximum likelihood estimation and online density estimation. We apply this framework to X-ray computed tomography reconstruction by explicitly modeling the non-linear physics of photon attenuation and Poisson noise, thereby bridging the gap between rigorous sequential statistics and modern deep learning-based reconstruction. Further related are also recent works that apply conformal prediction to inverse problems (Kutiel et al., 2023; Ekmekci & Cetin, 2025); Amougou et al. (2025) extend self-supervised conformal prediction to Poisson imaging via an unbiased risk estimator, requiring no labelled calibration data. However, standard marginal coverage in conformal prediction holds only on average over the population, not for specific objects.

## 2. Setting

Computed tomography is inherently volumetric, with the goal to obtain 3D tomographic reconstructions from 2-dimensional projection data. For simplicity, we focus on the mathematically equivalent setting of reconstructing 2D cross-sections from 1-dimensional projections. This significantly lowers computational demands, facilitates rapid experimentation and extensive benchmarking. The proposed framework extends to 3D volumes by applying the confidence estimation method to individual axial slices (in parallel-beam geometry), or directly to volumetric data.

Mathematically, we represent 2-dimensional tomographic slices as monochromatic images $\mathbf{x} \in \mathcal{X} := [0,1]^{r \times r}$ with side-length $r \in \mathbb{N}$, representing attenuation coefficients on a discretized sample. We rescale the physical attenuation coefficients to the unit interval to ensure numerical stability and standardize the dynamic range.

### 2.1. Physical Forward Model and Likelihood

We assume a parallel beam geometry where measurements are collected at specific view angles ranging from 0 to 180 degrees. Let $\mathcal{A} := [0, 180)$ be the domain of valid view angles. For any angle $\alpha \in \mathcal{A}$, let $R_\alpha \in \{0, 1\}^{r \times r^2}$ be the Radon transform (Kak & Slaney, 2001), representing the sparse projection matrix encoding the geometry of the scanner, mapping the $r^2$ image pixels to $r$ detector channels. Let $I_0 \in (0, \infty)$ denote the incident X-ray intensity.

The mean photon counts are modeled according to the *Beer-Lambert law* (Hsieh, 2015), which describes the exponential attenuation of X-rays as they pass through matter. For any 2-dimensional slice $\mathbf{x} \in \mathcal{X}$, measurement angle $\alpha \in \mathcal{A}$, and incident beam intensity $I_0 \in (0, \infty)$, the expected photon count at detector bin $i \in \{1, \dots, r\}$ is:

$$\lambda_i(\mathbf{x}, \alpha, I_0) := I_0 \exp\left(-\frac{l}{r}[R_\alpha \mathbf{x}]_i\right), \quad (1)$$

where $[R_\alpha \mathbf{x}]_i$ is the discrete line integral (Radon transform) of the attenuation map along the $i$-th ray. The constant $l$ converts the dimensionless pixels into physical path lengths.

We model detector measurements (photon counts) as independent Poisson random variables with rate $\lambda_i(\mathbf{x}, \alpha, I_0)$. For any angle $\alpha \in \mathcal{A}$, incident intensity $I_0 \in (0, \infty)$ and observed counts $\mathbf{y} \in \mathbb{N}_0^r$, the likelihood is:

$$p_\mathbf{x}(\mathbf{y} \mid \alpha, I_0) := \prod_{i=1}^r \exp\left(-\lambda_i(\mathbf{x}, \alpha, I_0)\right) \frac{\lambda_i(\mathbf{x}, \alpha, I_0)^{y_i}}{y_i!}.$$

### 2.2. Sequential Data Acquisition Protocol

We primarily focus on a *sparse-view* acquisition protocol where measurements are acquired for a sequence of angles $(\alpha_t)_{t \in \mathbb{N}}$, one angle at a time, with fixed incident intensity $I_0$. This setting is particularly challenging because the number of unknowns $r^2$ exceeds the number of scalar measurements $t \cdot r$ during the initial phase. Consequently, the inverse problem is ill-posed: the likelihood function lacks a unique global maximizer, or exhibits a flat landscape where many distinct images are statistically consistent with the observed data. This regime serves as a challenging testbed for uncertainty estimation, enabling us to demonstrate that likelihood mixing, combined with data-driven priors, yields tight, informative uncertainty estimates, even when the likelihood landscape itself is flat.

Let $(\alpha_t)_{t \in \mathbb{N}}$ be a deterministic sequence of angles where for all $t \in \mathbb{N}$, $\alpha_t \in \mathcal{A}$. We note that while the proposed formalism naturally extends to stochastic and data-adaptive scan trajectories, we focus on the deterministic setting standard in clinical protocols. We assume a fixed source intensity $I_0 \in (0, \infty)$. At each step $t$, we observe photon counts $\mathbf{y}_t \in \mathbb{N}_0^r$ at the detector. The data observed at step $t$ is defined as:

$$Z_t = (\alpha_t, \mathbf{y}_t) \in \mathcal{A} \times \mathbb{N}_0^r,$$

where conditional on the angle $\alpha_t$, the counts follow the Poisson model $\mathbf{y}_t \sim p_{\mathbf{x}^*}(\cdot \mid \alpha_t, I_0)$ for a ground-truth attenuation image $\mathbf{x}^* \in \mathcal{X}$.

Complementary to sparse-view acquisition, Appendix E considers a *dense-view* acquisition protocol. In this setting, a full grid of uniform angles is acquired at each step $t$ using a sequence of exposure intensities $(I_{0,t})_{t \in \mathbb{N}}$. This parallel evaluation serves two purposes: it allows us to verify whether the method's performance is specific to the under-constrained nature of the sparse setting, and it tests the generalization of the framework to alternative acquisition protocols (e.g., for fast rotating scanners which are ill-suited for sparse acquisition).

# 3. Methodology

Our goal is to derive *finite-sample* and *anytime-valid* uncertainty estimates for the reconstructed tomographic image. Formally, we seek to construct a *confidence sequence* $(C_t)_{t \in \mathbb{N}}$: a sequence of subsets $C_t \subset \mathcal{X}$ of statistically plausible images (confidence regions) that shrinks as more data is collected, while guaranteeing that the ground truth $\mathbf{x}^* \in \mathcal{X}$ remains within the set with high probability at all times.

**Definition 3.1** (Confidence Sequence). A sequence of sets $(C_t)_{t \in \mathbb{N}}$ is a $(1 - \delta)$-confidence sequence for the ground truth $\mathbf{x}^*$ if

$$\mathbb{P}\left(\forall t \in \mathbb{N} : \mathbf{x}^* \in C_t\right) \geq 1 - \delta.$$

Unlike standard confidence intervals which are valid only at a fixed sample size $T \in \mathbb{N}$, a confidence sequence is required to contain the ground truth image at any time $t \in \mathbb{N}$. This property is beneficial for sequential imaging scenarios, as it ensures that the uncertainty quantification remains valid regardless of when the acquisition is terminated.

We emphasize that we work in a *frequentist* framework, in which probability is taken only over the randomness in the data. In particular, the ground truth $\mathbf{x}^*$ is fixed, and the probability statement quantifies the random evolution of the confidence sequence $(C_t)$, in contrast to Bayesian credible sets that assign probability mass directly to $\mathbf{x}^*$.

## 3.1. Sequential Likelihood Mixing

We derive confidence sequences for CT reconstruction based on the *sequential likelihood mixing* framework (Kirschner et al., 2025). Sequential likelihood mixing uses the martingale properties of likelihood ratios to define data-driven confidence bounds based on the accumulated evidence.

The geometry of the confidence sets is determined by the *negative log-likelihood* of the observed data sequence. For a sequence of measurements $Z_s = (\alpha_s, \mathbf{y}_s) \in \mathcal{A} \times \mathbb{N}_0^r$ at times $s = 1, \ldots, t$, the negative log-likelihood of a candidate image $\mathbf{x} \in \mathcal{X}$ is

$$L_t(\mathbf{x}) := -\sum_{s=1}^{t} \log p_{\mathbf{x}}(\mathbf{y}_s \mid \alpha_s, I_0).$$

We define the confidence set $C_t$ at step $t$ as the level set of the negative log-likelihood with a time-dependent threshold $\beta_t \in \mathbb{R}$ and failure probability $\delta \in [0, 1]$:

$$C_t := \left\{\mathbf{x} \in \mathcal{X} \mid L_t(\mathbf{x}) \leq \beta_t + \log \tfrac{1}{\delta}\right\}. \qquad (2)$$

The core challenge lies in choosing the *confidence coefficient* $\beta_t$ such that the coverage condition (Definition 3.1) holds, while making $\beta_t$ as small as possible, corresponding to tighter confidence regions.

The sequential likelihood mixing framework provides a constructive, data-driven solution to this dual problem — ensuring coverage while minimizing $\beta_t$ — by defining $\beta_t$ as the *sequential negative log-likelihood* for a sequence of reconstructions $\hat{\mathbf{x}}_0, \ldots, \hat{\mathbf{x}}_{t-1}$,

$$\beta_t = -\sum_{s=1}^{t} \log p_{\hat{\mathbf{x}}_{s-1}}(\mathbf{y}_s \mid \alpha_s, I_0). \qquad (3)$$

Intuitively, $\beta_t$ corresponds to a sequential *testing error*, measured by the negative log-likelihood of the data observation $\mathbf{y}_s$, conditioned on measurement angle $\alpha_s$ under the prediction $\hat{\mathbf{x}}_{s-1}$ from the previous time step. The magnitude of the threshold, and thereby the tightness of the confidence region, is determined by the ability of the reconstruction $\hat{\mathbf{x}}_{s-1}$ to faithfully predict the subsequent measurement $\mathbf{y}_s$. Importantly, the construction is valid for *any* sequence of predictions $\hat{\mathbf{x}}_s$ (where $\hat{\mathbf{x}}_s$ is formally required to make use of data only up to step $s$), including neural-network based reconstructions.

Moreover, by introducing the notion of a *mixing distribution* $\mu_s$ over the image space $\mathcal{X}$, the construction extends to probabilistic estimators and generative models (e.g., diffusion models). The following proposition specializes Theorem 3 of Kirschner et al. (2025) to the CT setting.

**Proposition 3.2** (Sequential Likelihood Mixing for CT). *Let $(\mu_s)_{s \in \mathbb{N}_0}$ be a sequence of distributions on $\mathcal{X}$, where for each $s \in \mathbb{N}$, $\mu_s$ depends only on data observed up to step $s$. For any error level $\delta \in (0, 1)$, define the threshold*

$$\beta_t := -\sum_{s=1}^{t} \log \int_{\mathcal{X}} p_{\mathbf{z}}(Z_s) \, d\mu_{s-1}(\mathbf{z}).$$

*Then $(C_t)_{t=1}^{\infty}$ with $C_t := \left\{\mathbf{x} \in \mathcal{X} \mid L_t(\mathbf{x}) \leq \beta_t + \log \tfrac{1}{\delta}\right\}$ for all $t \in \mathbb{N}$ is a $(1 - \delta)$-confidence sequence for $\mathbf{x}^* \in \mathcal{X}$, i.e., $\mathbb{P}(\forall t \in \mathbb{N} : \mathbf{x}^* \in C_t) \geq 1 - \delta$.*

The result holds for any sequence of mixing distributions, and formally only requires realizability, i.e., $\mathbf{x}^* \in \mathcal{X}$ and the observation data $\mathbf{y}_t$ is sampled from $p_{\mathbf{x}^*}(\cdot \mid \alpha_t, I_0)$[2]. Both assumptions are satisfied for the standard CT model in Section 2.1. The proof follows from observing that the sequential log-likelihood ratio is a martingale and applying Ville's inequality, and is given for completeness in Appendix A.

The definition of $\beta_t$ using mixing distributions recovers the special case in Equation (3) for a sequence of point predictions $\hat{\mathbf{x}}_t$ by setting $\mu_t = \delta_{\hat{\mathbf{x}}_t}$ as a Dirac measure. We refer to Kirschner et al. (2025) for further examples on how to instantiate confidence sequences, including Bayesian posteriors, variational inference and maximum likelihood estimation.

---

[2]The result further extends to adaptive data sequences, e.g., in sequential experimental design or active learning, and the realizability assumption can be relaxed (c.f., Kirschner et al., 2025).

In the context of this work, we explore U-Net ensembles and diffusion models as mixing distributions (Section 4.2).

We remark that prior works often present a formulation that analytically bounds $\beta_t$ via the cumulative regret of the mixing sequence relative to the maximum likelihood estimator (Abbasi-Yadkori et al., 2012; Lee et al., 2024b; Kirschner et al., 2025; Clerico et al., 2025). Formally, define the maximum likelihood estimate $\hat{\mathbf{x}}_t = \arg\min_{\mathbf{x} \in \mathcal{X}} L_t(\mathbf{x}_t)$, and replace $\beta_t$ with $L_t(\hat{\mathbf{x}}_t) + R_t$, using a regret bound $\beta_t - L_t(\hat{\mathbf{x}}_t) \leq R_t$. This results in a valid confidence set with a potentially looser threshold $\beta_t \leq L_t(\hat{\mathbf{x}}_t) + R_t$. Using the empirical $\beta_t$ directly is therefore always at least as tight. Moreover, valid regret bounds are unavailable for deep learning reconstruction methods, making the empirical formulation the more practical choice in our setting.

While the sets $C_t$ are mathematically defined by Equation (2) as level sets of negative log-likelihood, they serve as the foundation for concrete, practical diagnostic tools. We utilize the derived bounds to construct interpretable pixel-wise uncertainty visualizations (Section 4.6), detect geometric misalignment such as rotations (Section 4.5), and identify model hallucinations (Section 4.7) by flagging instances where the reconstruction is statistically inconsistent with the measurements.

From a computational perspective, at each time step the threshold $\beta_t$ is computed by sequentially evaluating the negative log-likelihood of the observed data. Given $\beta_t$, checking whether a test point $\mathbf{x}$ belongs to the confidence set $C_t$ requires another evaluation of the negative log-likelihood $L_t(\mathbf{x})$. On the other hand, directly evaluating the high-dimensional confidence region $C_t$ can be challenging. One possibility is to use the defining Equation (2) as an optimization constraint, allowing efficient computation of representative points within the confidence set (see Section 4.6). In most cases, the dominant cost is the predictor itself, for example, diffusion posterior sampling requires evaluating the likelihood for guidance; whereas computing $\beta_t$ requires only evaluating the sequential likelihood, which is substantially cheaper than the sampling itself. For 3D volumes, the approach scales linearly in the number of axial slices under parallel-beam geometry, as each slice can be processed independently.

### 3.2. Uncertainty Estimates for Generative Models

The power of Proposition 3.2 lies in the flexibility of choosing the mixing distributions $\mu_s$. One possible choice is to set $\mu_s$ as a Bayesian posterior (c.f., Kirschner et al., 2025), however, the key questions of choosing an informative prior and approximating the posterior distribution remain. Modern generative models are pre-trained on prior data to capture domain-specific structure and enable efficient sampling of the data distribution conditioned on observational data.

Specifically, for any step $s \in \mathbb{N}$, assume a generative model $\mathcal{M}$ produces $K \in \mathbb{N}$ samples $\tilde{\mathbf{x}}_s^{(1)}, \ldots, \tilde{\mathbf{x}}_s^{(K)} \in \mathcal{X}$, conditioned on observation data up to time $s$. We then define $\mu_s$ as a uniform mixture of Diracs centered on these estimates, $\mu_s = \frac{1}{K} \sum_{k=1}^{K} \delta_{\tilde{\mathbf{x}}_s^{(k)}}$. Substituting the discrete mixture into Proposition 3.2 leads to the following confidence coefficient:

$$\beta_t := -\sum_{s=1}^{t} \log \frac{1}{K} \sum_{k=1}^{K} p_{\tilde{\mathbf{x}}_{s-1}^{(k)}}(\mathbf{y}_s \mid \alpha_s). \qquad (4)$$

In Section 4, we instantiate this approach for various mixing distributions, ranging from classical single-point estimates to diffusion models and deep multi-point ensembles. We note that recent work has shown that plug-and-play diffusion samplers exhibit a growing gap to the true Bayesian posterior under sparse measurements (Moroy et al., 2026); crucially, sequential likelihood mixing yields valid confidence sets for *any* mixing distribution, and posterior accuracy is not required for validity.

### 3.3. Ill-Posedness and Model Misspecification

Because $C_t$ is a level set of the likelihood function, it inherits any identifiability limitations of the forward model. In the sparse-view regime, where the number of unknowns exceeds the number of measurements, the likelihood may be flat over a large region, and the resulting $C_t$ can cover a large portion of the image space. Restricting to a bounded image space $\mathcal{X} = [0, 1]^{r \times r}$ guarantees that $C_t$ always has *finite* volume, however does not directly address the identifiability problem. Two complementary strategies can tighten $C_t$ in ill-posed regimes: (i) imposing additional structural constraints on $\mathcal{X}$, such as total-variation bounds or smoothness priors, intersected with the likelihood level set; and (ii) imposing learned structure, e.g. via using Diffusion $C_t$-Boundary Sampling (Section 4.6) to project the confidence set onto the manifold of realistic images learned by a diffusion prior.

The construction further assumes a well-specified forward model (*realizability*) with the ground truth $\mathbf{x}^* \in \mathcal{X}$ and the observed data generated from $p_{\mathbf{x}^*}$. Real CT acquisitions involve additional physical effects — such as beam hardening, scattering, or geometric calibration errors — that are not captured by the monochromatic Beer–Lambert / Poisson model. Misspecification can be mitigated by choosing a forward model that accounts for these effects (e.g., polychromatic attenuation or additive electronic noise). Formal robustness results for certain classes of misspecification, including sub-Gaussian deviations from the nominal likelihood, are established in Kirschner et al. (2025, Appendix D). Under severe misspecification the type-I error guarantee may not hold, practitioners should verify model adequacy (e.g., via residual analysis on held-out projections).

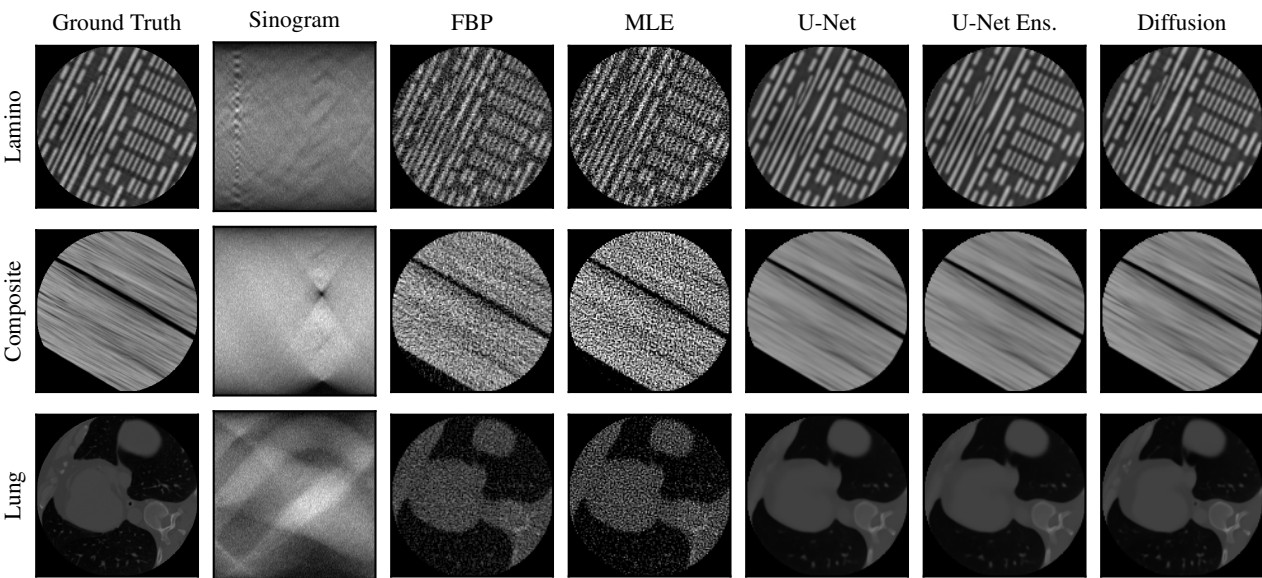

*Figure 1.* Ground truth samples, measurement data (sinogram) and final reconstructions at total intensity $I_{\text{total}} = 10^7$. While classical algorithms (FBP, MLE) exhibit significant noise and artifacts, the deep learning predictors (U-Net, U-Net Ensemble, Diffusion) successfully recover fine structural details and sharp edges.

## 4. Experimental Results

We empirically evaluate the performance of likelihood-based confidence sets on three distinct tomographic datasets for medical, industrial and scientific imaging tasks.

### 4.1. Datasets

We use three diverse tomographic datasets, representing medical, industrial, and materials science imaging scenarios, visualized in Figure 1. First, the **Lamino (Industrial)** dataset contains Ptychographic X-ray Laminography (PyXL) scans of an integrated circuit, acquired at the Swiss Light Source (Holler et al., 2019). These images are characterized by highly regular, dense geometric structures with sharp edges and Manhattan-like geometries, typical of semiconductor manufacturing.

Second, the **Composite (Materials)** dataset consists of X-ray CT scans of non-crimp fabric reinforced composites (Auenhammer et al., 2020). These images display a structured morphology where fiber bundles are primarily aligned in one direction, presenting a challenging reconstruction target with high-frequency details and low contrast.

Finally, the **Lung (Medical)** dataset is sourced from the *Lung Image Database Consortium* (LIDC-IDRI) collection (Armato et al., 2011). These medical scans feature soft tissue structures, high-contrast air cavities, and potential pathologies such as nodules. For our experiments, we utilize a subset of 2D axial slices rescaled to $r \times r$ resolution.

### 4.2. Reconstruction Methods and Mixing Strategies

We instantiate the likelihood mixing framework for several reconstruction methods, including point estimates ($K = 1$) via Equation (3), and multi-point estimates ($K > 1$) via Equation (4), ranging from classical reconstruction algorithms to deep generative models.

As **single-point baselines** ($K = 1$), we evaluate *Filtered Backprojection* (FBP), centering $\mu_{s-1}$ on the standard analytic reconstruction, and *Maximum Likelihood Estimation* (MLE), where the center is obtained via gradient-based minimization of the data likelihood (with early stopping). We contrast these with a standard denoising U-Net, which provides a single-point estimate directly from FBP inputs.

Second, we obtain **mixing distributions** ($K > 1$) that capture epistemic uncertainty, using two strategies: a U-Net Ensemble with 10 members trained with different random seeds, and a Diffusion model generating diverse samples via conditional diffusion with soft data-consistency guidance (Chung et al., 2023; Barba et al., 2024). Full training and architecture details are provided in Appendix B. To better understand the effect of mixing, we further compare to a **mean-aggregation strategy**, where the multi-point estimates are averaged into a single point-prediction $\bar{\mathbf{x}}_{s-1} = \frac{1}{K} \sum_{k=1}^{K} \hat{\mathbf{x}}_{s-1}^{(k)}$.

### 4.3. Evaluation Protocol

We generate observation data for each dataset by simulating the forward process (Equation (1)) with a Poisson likelihood.

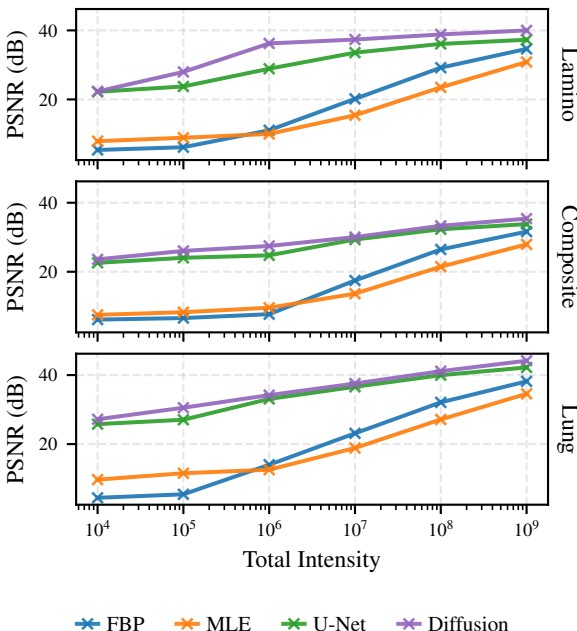

*Figure 2.* PSNR of the final reconstruction vs. total intensity. The (imperceptibly thin) shaded regions indicate mean $\pm$ SEM (100 test set images, 10 seeds).

The CT inverse problem is known to be ill-conditioned, and generating data using the exact same numerical model used for reconstruction can lead to overly optimistic performance (the so-called "inverse crime", see Wirgin, 2004; Mueller & Siltanen, 2012). Therefore we compute the observation data at a higher resolution ($r = 256$), whereas all reconstruction methods use down-sampled data at a detector resolution $r = 128$ and image size $r \times r = 128 \times 128$.

We evaluate on a held-out test set of 100 images from each of the three datasets (Lamino, Composite, Lung). To account for the stochastic nature of photon emission, we conduct 10 independent trials for each image using distinct random seeds for the Poisson noise generation. The seeds are shared across methods such that all confidence sets are for exactly the same observation sequences.

We assess the confidence sequences over a range of total intensities, defined as $I_{\text{total}} := t_{\text{final}} \cdot r \cdot I_0$, varying from $10^4$ to $10^9$. We simulate a total acquisition of 200 view angles. We reserve the first 10 projection angles to compute an initial reconstruction, which serves as the initialization for the mixing distribution $\mu_0$. The confidence sequence is then computed over the remaining horizon of $t_{\text{final}} = 190$ steps, processing view angles 11 to 200. Because $\mu_0$ is a fixed prior to observing the data used in the likelihood ratio, the theoretical validity of the confidence sequence is strictly preserved.

## 4.4. Reconstruction Quality

We first verify the reconstruction quality of the baselines and neural-network reconstructions. Figure 2 quantifies reconstruction quality in terms of Peak Signal-to-Noise Ratio (PSNR) across all datasets.

Unsurprisingly, the results reveal a clear hierarchy in terms of image quality. The diffusion-based predictor achieves the highest average PSNR across all intensity levels. The U-Net reconstruction follows closely, trailing the diffusion model by approximately 2 dB (the near-identical U-Net Ensemble is omitted here; see Section 4.2). In contrast, the analytical and iterative baselines (FBP, MLE) yield significantly lower fidelity reconstructions, particularly in the low-dose regime. Visual comparisons in Figure 1 confirm that while classical methods struggle with noise and artifacts, neural models successfully recover fine structural details.

However, we emphasize that high reconstruction fidelity (PSNR) does not imply a superior performance for sequential likelihood mixing since a higher PSNR score does not imply a lower sequential negative log-likelihood.

## 4.5. Tightness of Confidence Sets

We assess the tightness of $C_t$ by comparing $\beta_t$ for different reconstruction methods. Since $C_t$ is a level set of the negative log-likelihood, the size of the confidence set directly scales with $\beta_t$ (i.e., $\beta_t \leq \beta'_t$ implies $C_t(\beta_t) \subset C_t(\beta'_t)$). We focus on the confidence sets $C_{t_{\text{final}}}$ at time step $t_{\text{final}} = 190$, which we will assume for the remainder of this section.

**Effect of Mixing.** For multi-output predictors (U-Net Ensemble and Diffusion), we investigate the effect of using mixing distribution $\mu_s$ (Equation (4)), compared to a *mean* approach (a single point mass centered at the average prediction). Figure 6 in Appendix D.1 shows the difference $\beta_{t_{\text{final}}}^{\text{mean}} - \beta_{t_{\text{final}}}^{\text{mix}}$. We observe that the mixture strategy consistently yields a lower (better) sequential negative log-likelihood, with increasing gap at higher intensity. This advantage stems from the robustness of the mixture likelihood. Even if several predictions in the ensemble have low likelihood for the observed measurement, a single accurate prediction can dominate the log-sum-exp, preserving a low sequential negative log-likelihood. In contrast, while mean prediction typically achieves lower square-error (or higher PSNR), if the average image is inaccurate due to smoothing artifacts, the resulting log-likelihood penalty is more severe. Consequently, given its superior performance, we restrict the remainder of our evaluation to the mixture strategy.

**Confidence Coefficient Comparison.** Figure 3 illustrates the evolution of the gap $\beta_{t_{\text{final}}} - L_{t_{\text{final}}}(\mathbf{x}^*)$ across different methods. We observe a distinct ordering in performance. Analytical baselines like FBP yield significantly looser bounds, with the sequential negative log-likelihood

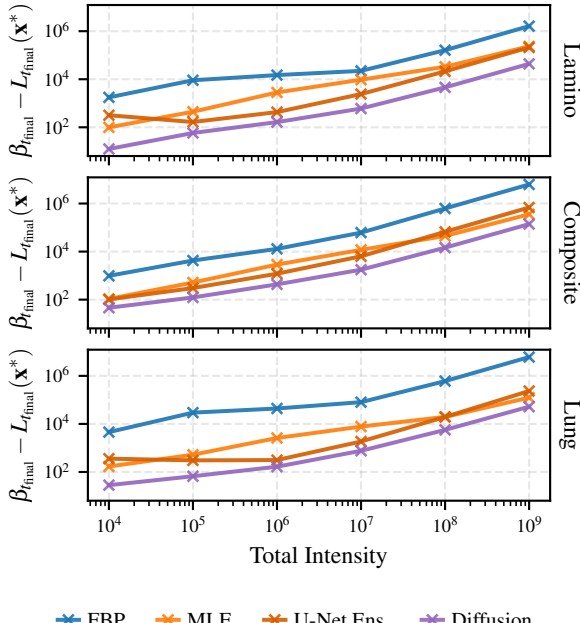

*Figure 3.* Difference between sequential negative log-likelihood and ground truth image negative log-likelihood $\beta_{t_{\text{final}}} - L_{t_{\text{final}}}(\mathbf{x}^*)$. Shaded regions indicate $\pm$ SEM (100 test set images, 10 seeds).

gap exceeding $10^6$ at high intensities. While iterative methods (MLE) and the U-Net Ensemble reduce this gap, the diffusion-based mixing consistently achieves the lowest sequential negative log-likelihood across all datasets, resulting in the smallest confidence set $C_t$. For example, on the Lamino dataset at total intensity $10^9$, the diffusion model achieves an average gap of approximately $4.4 \times 10^4$, an order of magnitude tighter than MLE ($\approx 2.3 \times 10^5$) and substantially lower than FBP ($\approx 1.6 \times 10^6$). A U-Net point predictor (not shown) achieves nearly identical performance compared to the U-Net ensemble mixing in our experiments.

**Empirical Validity.** While Proposition 3.2 guarantees statistical validity, we investigate how close the empirical error rate is to the theoretical worst-case failure probability $\delta$. We track the *crossover rate*, defined as the empirical frequency with which the ground truth negative log-likelihood exceeds the threshold $\beta_t + \log \frac{1}{\delta}$ for at least one step $t \in \{1, \ldots, 190\}$, corresponding to a type-I error. Figures 4 and 7 confirm that all methods strictly satisfy the coverage guarantee (error rate $\leq \delta = 0.05$). Notably, at low total intensities, the diffusion model approaches the nominal 5% error rate (e.g., $\approx 3\%$ on Composite), suggesting that the derived bounds are non-conservative and efficient in the low-dose regime. The relationship between $\delta$ and empirical coverage is analyzed in more detail in Appendix D.2.

**Geometric Specificity.** Finally, we assess the diagnostic capability of the confidence sets by testing their sensitivity

to geometric perturbations. We rotate the ground truth $\mathbf{x}^*$ and measure the exclusion rate, corresponding to the type-II error rate, or the power of the statistical test $\mathbf{x}^*_{rot} \notin C_t$. As shown in Figure 4, diffusion-based confidence sets exhibit high specificity; rotated images are excluded with high probability even for small angles ($1°$) at high intensity. Conversely, looser baselines like FBP are unable to exclude rotated copies even at significant angles ($> 4°$), highlighting that tighter, data-driven bounds provide a more sensitive mechanism for detecting reconstruction errors. We present the results for the Lung and Composite datasets in Figure 8.

### 4.6. Uncertainty Visualization

While the confidence set $C_t$ provides a statistically valid confidence region, visualization of the uncertainty requires projecting high-dimensional sets onto the image coordinates. We explore two distinct approaches to constructing pixel-wise uncertainty intervals: a worst-case optimization approach, and a data-driven Diffusion sampling strategy.

We compare two strategies for projecting the high-dimensional set $C_t$ onto pixel-wise intervals. First, **worst-case optimization** approximates independent pixel-wise extrema via projected gradient descent subject to the constraint $\mathbf{x} \in C_t$. This yields conservative bounds by attempting to explore the full extent of the feasible set. In particular, the definition of $C_t$ does not enforce any structure on the set of images $\mathcal{X}$. Second, to address this limitation, we propose **Diffusion $C_t$-Boundary sampling** to obtain tighter, physically plausible intervals. To this end, we generate diverse samples on the boundary of $C_t$ using a diffusion process guided by a diversity objective $\mathcal{L}_t$, enforcing the likelihood constraint while maximizing diversity (see Appendix C.2). We then compute pixel-wise Student-t intervals from these samples.

To contextualize the performance of the likelihood-based methods, we compare against five baselines drawn from the literature on posterior sampling and uncertainty quantification for inverse problems. **Equivariant Bootstrap** (Pereyra & Tachella, 2024) augments the FBP–U-Net estimator with random in-plane rotations of the prior reconstruction before re-simulating Poisson counts, producing 1000 replicates. **SK-ROCK** (Abdulle et al., 2018; Pereyra et al., 2020) is an accelerated Langevin sampler. We apply it at the final acquisition step $t_{\text{final}}$ with target potential $L_{t_{\text{final}}}(\mathbf{x}) + g^\tau(\mathbf{x})$, where $g^\tau$ is the Moreau–Yosida envelope (with smoothing parameter $\tau > 0$) of a total-variation prior; we keep 1000 post-burn-in samples from a single chain. **FBP Bootstrap** and **U-Net Bootstrap** are non-parametric bootstraps: we generate 1000 datasets by resampling angle indices with replacement from the observed $(\alpha_t, \mathbf{y}_t)$ pairs and reconstruct each resample with the corresponding base estimator. From the first four— all sampling-based—we construct pixel-wise percentile in-

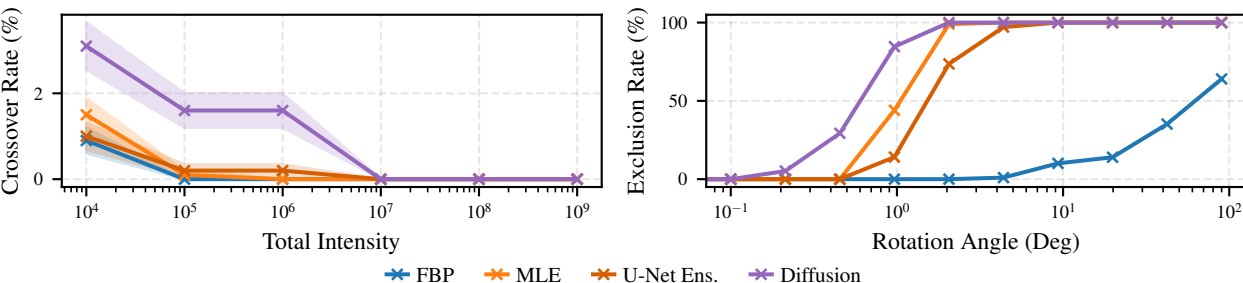

*Figure 4.* Crossover and exclusion rates vs. total intensity for Lamino dataset and error level $\delta = 0.05$, corresponding to type-I and type-II error rates. Crossover rate: rate at which the ground truth image is not inside all confidence sets of the confidence sequence. Exclusion rate: rate at which the rotated ground truth image is not included in the last confidence set. Shaded regions indicate mean $\pm$ SEM (100 test set images, 10 seeds).

tervals at level $\delta = 0.05$; the fifth, the **U-Net Ensemble** from Section 4.2, instead provides Student-$t$ intervals over its 10 independent networks. The width-controlling hyper-parameter of Equivariant Bootstrap and SK-ROCK is calibrated per (dataset, total intensity) on the first 10 test images, disjoint from the held-out evaluation set; full procedure and hyperparameters are deferred to Appendix D.4.

**Results.** Figure 5 (Appendix C) reports pixel-wise coverage and confidence-interval width on the held-out test set of 100 images for each dataset. Unlike the reconstruction benchmarks, these experiments use a single noise seed per image. The methods split into two regimes. Worst-Case, FBP Bootstrap, SK-ROCK, and Diffusion $C_t$-Boundary Sampling all reach the nominal $1-\delta = 0.95$ coverage target across datasets and intensities. Among these four, Diffusion $C_t$-Boundary Sampling consistently produces the narrowest intervals across the full intensity range — substantially tighter than Worst-Case and FBP Bootstrap, and tighter than SK-ROCK at low and intermediate intensities; only in the high-dose regime do SK-ROCK and Boundary converge to comparable widths. The remaining methods — Equivariant Bootstrap, U-Net Bootstrap, and U-Net Ensemble — produce visibly tighter intervals but systematically miss the coverage target, with Equivariant Bootstrap undershooting most severely despite per-cell hyperparameter calibration. This suggests that rotation-augmented bootstrapping and ensemble disagreement underestimate the posterior spread on these datasets. The contrast with FBP Bootstrap is informative: a non-parametric bootstrap of the same FBP estimator achieves nominal coverage where its U-Net-based counterpart does not, indicating that the failure is specific to the learned estimator rather than to the bootstrap principle. Taken together, Diffusion $C_t$-Boundary Sampling is the only method that achieves nominal coverage *and* the tightest intervals across the full operating range, establishing it as the strongest pixel-wise UQ method in this comparison. Example uncertainty visualizations are shown in Figure 17.

### 4.7. Hallucination detection

As an illustrating application, we make use of likelihood-based confidence sets to detect hallucinations — plausible looking, but incorrect reconstructions. To this end, we sample reconstructions $\tilde{\mathbf{x}}_t^{(1)}, \ldots, \tilde{\mathbf{x}}_t^{(K)}$ from the Diffusion model, and check if $\tilde{\mathbf{x}}_t^{(k)} \in C_t$ at each acquisition step $t$. We then compute PSNR reconstruction quality conditioned on plausible samples ($\tilde{\mathbf{x}}_t^{(k)} \in C_t$) and hallucinations (defined as $\tilde{\mathbf{x}}_t^{(k)} \notin C_t$). Figure 16 in Appendix E.4 shows that plausible samples achieve significantly higher PSNR scores, demonstrating that the confidence set correctly identifies inconsistent generations. We refer to Appendix E.4 for details.

## 5. Conclusion

We presented a framework for uncertainty quantification in computed tomography based on sequential likelihood mixing. We empirically demonstrated that modern deep learning priors — specifically U-Nets, deep ensembles, and diffusion models — can be used to construct tight, data-driven confidence regions with finite-sample, anytime validity guarantees for a realistic physical forward model of X-ray tomography and Poisson detector noise.

Beyond theoretical guarantees, we highlighted the practical utility of likelihood-based confidence regions. We demonstrated that these confidence sets serve as an effective mechanism for hallucination detection, allowing practitioners to flag structurally plausible reconstructions that are nonetheless statistically inconsistent with the observed measurement data. Further, we proposed visualization techniques to project these high-dimensional sets into interpretable pixel-wise uncertainty maps. Collectively, these contributions bridge the gap between the raw performance of deep learning and the safety-critical requirements of medical imaging, offering a path toward trustworthy, uncertainty-aware computed tomography.

# Acknowledgements

The authors thank Jonas Peters and Luis Barba for helpful discussions. The project received funding support from the CHIP project of the SDSC under grant no. C22-11L.

# Impact Statement

This paper presents foundational research on statistical uncertainty quantification for computed tomography, and as such, there are no direct potential ethical or societal consequences that we foresee. As any statistical approach, the methodology presented here is subject to modeling assumptions, which, if violated, could lead to a misrepresentation of uncertainty. Further validation is needed before deploying the proposed method in high stakes settings such as medical analysis.

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

# A. Proof of Proposition 3.2

For completeness, we reproduce the proof of Proposition 3.2, following Kirschner et al. (Theorem 3, 2025).

Formally, let $\mathcal{F}_t$ be the data filtration generated by the observations $(Z_1, \ldots, Z_t)$. Let $(\mu_t)_{t \in \mathbb{N}_0}$ be an $\mathcal{F}_t$-adapted sequence of mixing distributions over $\mathcal{X}$, and $\mu_0$ a prior mixing distribution that is chosen before any data is observed. The sequential marginal likelihood ratio is defined as follows for any $\mathbf{x} \in \mathcal{X}$,

$$S_t(\mathbf{x}) = \prod_{s=1}^{t} \frac{\int p_{\mathbf{x}'}(Z_s) d\mu_{s-1}(\mathbf{x}')}{p_{\mathbf{x}}(Z_s)} \,.$$

An application of Fubini's theorem implies that $S_t(\mathbf{x}^*)$ is a non-negative martingale under $\mathcal{F}_t$ with $\mathbb{E}[S_0(\mathbf{x}^*)] = 1$,

$$\mathbb{E}[S_t(\mathbf{x}^*) \mid \mathcal{F}_t] \stackrel{(1)}{=} S_{t-1} \cdot \mathbb{E}\left[\int \frac{\int p_{\mathbf{x}'}(Z_t) d\mu_{t-1}(\mathbf{x}')}{p_{\mathbf{x}}(Z_t)} \mid \mathcal{F}_t\right]$$

$$\stackrel{(2)}{=} S_{t-1} \cdot \int \frac{\int p_{\mathbf{x}'}(z_t) d\mu_{t-1}(\mathbf{x}')}{p_{\mathbf{x}}(z_t)} p_{\mathbf{x}}(z_t) dz_t$$

$$\stackrel{(3)}{=} S_{t-1} \cdot \int \int \frac{p_{\mathbf{x}'}(z_t)}{p_{\mathbf{x}}(z_t)} p_{\mathbf{x}}(z_t) dz_t d\mu_{t-1}(\mathbf{x}')$$

$$= S_{t-1} \cdot \int \int p_{\mathbf{x}'}(z_t) dz_t d\mu_{t-1}(\mathbf{x}')$$

$$\stackrel{(4)}{=} S_{t-1} \cdot \int 1 \, d\mu_{t-1}(\mathbf{x}') = S_{t-1} \,.$$

Here, (1) follows from the conditioning on $\mathcal{F}_t$ and the definition of $S_t$, (2) uses realizability of the observation likelihood, (3) is a consequence of Fubini's theorem, and (4) follows from the fact that probability densities integrate to one.

Using a martingale version of Markov's inequality known as Ville's inequality (Ville, 1939), it follows that

$$\mathbb{P}\left[\sup_{t \geq 0} S_t(\mathbf{x}^*) \geq \frac{1}{\delta}\right] \leq \delta \,.$$

Proposition 3.2 follows by rewriting the definition of $S_t$.

# B. Reconstruction Methods

## B.1. Classical Baselines

We utilize two classical reconstruction methods. First, we employ FBP with a standard Ramp filter, which serves as a fast, analytical reconstruction method. Second, we evaluate an approximate MLE. Rather than computing the exact maximizer, we approximate the MLE by performing a limited number of gradient updates on the negative log-likelihood objective. We initialize with the FBP reconstruction and perform 100 steps of Adam optimization with a learning rate of $10^{-2}$.

## B.2. Deep Learning Predictors

We implement both deterministic and stochastic deep learning predictors. All deep models utilize a holdout validation set for optimal checkpoint selection (early stopping) during training.

**Data Normalization and Conditioning.** For all deep learning models, input images (FBP reconstructions) are normalized to the range $[-1, 1]$. Similarly, the scalar conditioning variables—total intensity and number of angles—are linearly scaled to the $[-1, 1]$ range before being injected into the networks.

**U-Net and Ensembles.** We employ Transformer-U-Net hybrids (Vaswani et al., 2017; Chen et al., 2021; 2024) trained to map noisy FBP reconstructions to clean images. The network is conditioned on the normalized total intensity and number of angles. For the U-Net Ensemble, we train multiple independent instances of this architecture on the same training set, differing only in their random initialization seeds. During inference, the mixing distribution is constructed from the predictions of these individual members.

**Diffusion.** We train a conditional diffusion model (Ho et al., 2020) using the same architecture and conditioning scheme (FBP image, normalized intensity, normalized angles) as the U-Net. During the sequential inference phase, we construct the mixing distribution $\mu_{s-1}$ by generating a set of $K = 16$ replicates. We apply data-consistency guidance to ensure the samples align with the measurements. Specifically, we perform 20 gradient guidance steps. To stabilize the sampling trajectory, we linearly anneal the guidance learning rate from $10^{-2}$ to 0 over the course of the reverse diffusion process.

# C. Visualization and Optimization Details

In this appendix, we provide the implementation details for the uncertainty visualization methods described in Section 4.6.

## C.1. Worst-Case Optimization Strategy

To approximate the pixel-wise extrema of the confidence set $C_t$, we employ the following strategy. For a given input prediction $\hat{\mathbf{x}}$, we initialize $K = 8$ replicates $\{\mathbf{z}^{(k)}\}_{k=1}^{K}$ by cloning $\hat{\mathbf{x}}$ and adding small Gaussian noise ($\sigma = 10^{-3}$) to break symmetry. We then iteratively maximize the spread of these replicates while constraining them to remain within the confidence set $C_t$.

The optimization runs for a maximum of 1000 steps (processed in batches of 10 images). For each step, we perform

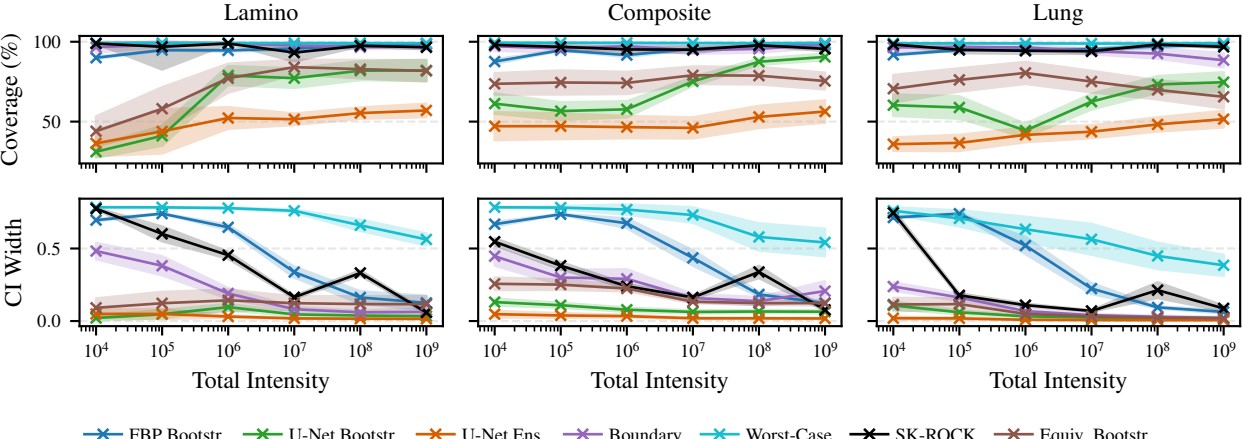

*Figure 5.* Pixel-wise coverage and confidence interval widths at $\delta = 0.05$. Diffusion $C_t$-Boundary Sampling and SK-ROCK remain at nominal coverage with substantially narrower intervals than Worst-Case and FBP Bootstrap; Equivariant Bootstrap, U-Net Bootstrap, and U-Net Ensemble produce tighter intervals but systematically undershoot the target. Shaded regions indicate mean $\pm$ standard deviation (100 test set images, single seed).

the following updates:

1. **Expansion Gradient:** We calculate the gradient to push each replicate away from the current ensemble mean $\bar{\mathbf{z}} = \frac{1}{K} \sum_k \mathbf{z}^{(k)}$. The raw gradient for replicate $k$ is $\mathbf{g}^{(k)} = \mathbf{z}^{(k)} - \bar{\mathbf{z}}$.

2. **Physical Constraint Masking:** To prevent wasted optimization effort at physical boundaries, we zero out gradients for pixels that have already reached the image bounds. Specifically, if a pixel $z_{i,j}^{(k)} > 0.999$ and the gradient is positive, or $z_{i,j}^{(k)} < 0.001$ and the gradient is negative, the gradient at that position is set to 0. The remaining gradients are normalized to unit length.

3. **Update and Projection:** We update the replicates using the masked gradients with an initial step size $\eta = 2.0$. Immediately after the update, we project any violator back into the confidence set $C_t$. The projection is solved via an inner optimization loop of up to 10,000 steps using Adam (learning rate $10^{-2}$), which minimizes the cumulative negative log-likelihood until the constraint $L_t(\mathbf{z}^{(k)}) \leq \beta_t + \log \frac{1}{\delta}$ is satisfied.

We employ a dual learning rate adaptation strategy. First, if the inner projection step becomes difficult (requiring $> 10$ iterations), the step size $\eta$ is decayed by a factor of 0.9. Second, if the mean spread objective plateaus (improvement $\leq 10^{-5}$), we aggressively decay $\eta$ by a factor of 0.5 and increment a patience counter. The process terminates early if the patience counter exceeds 10. The final uncertainty interval for a pixel is defined by the minimum and maximum values observed across all $K$ optimized replicates.

### C.2. Diffusion $C_t$-Boundary Sampling

We utilize a conditional diffusion model to sample images that lie on the boundary of the confidence set $C_t$. We employ a standard reverse diffusion process. At each timestep $\tau$ of the reverse process, we first estimate the clean image $\hat{\mathbf{x}}_0(\mathbf{x}_\tau)$ from the current noisy state $\mathbf{x}_\tau$ using Tweedie's formula.

Before proceeding to the next diffusion step $\tau - 1$, we refine the estimates to ensure they cover the extent of the confidence set. Let $\{\hat{\mathbf{x}}_0^{(k)}\}_{k=1}^K$ be the estimated clean images for a batch of $K$ parallel diffusion chains. We perform 20 steps of gradient descent on these estimates using a learning rate of 0.01 and following loss function:

$$\mathcal{L}_t(\hat{\mathbf{x}}_0^{(1)}, \ldots, \hat{\mathbf{x}}_0^{(K)}) = \frac{1}{K} \sum_{k=1}^K \mathcal{L}_t' \left( \hat{\mathbf{x}}_0^{(k)}, \bar{\mathbf{x}}_0 \right),$$

where $\bar{\mathbf{x}}_0 = \frac{1}{K} \sum_{j=1}^K \hat{\mathbf{x}}_0^{(j)}$ is the batch mean. The per-sample loss $\mathcal{L}_t'$ is defined as:

$$\mathcal{L}_t'(\mathbf{x}, \bar{\mathbf{x}}) = \begin{cases} -\gamma \cdot \|\mathbf{x} - \bar{\mathbf{x}}\|_2^2 & \text{if } L_t(\mathbf{x}) \leq \beta_{t,\delta}, \\ L_t(\mathbf{x}) & \text{otherwise.} \end{cases}$$

Here, $\gamma = 1000$ is a diversity weight, and $\beta_{t,\delta} = \beta_t + \log \frac{1}{\delta}$. The first case penalizes samples that cluster around the mean, forcing them apart to explore the boundary of the set. The second case is active only when the sample falls outside the confidence set, providing a strong gradient signal to push it back into the valid region $C_t$.

After optimizing the estimates $\{\hat{\mathbf{x}}_0^{(k)}\}$, we re-inject the noise corresponding to timestep $\tau$ and proceed with the next reverse diffusion step. This guidance strategy ensures that

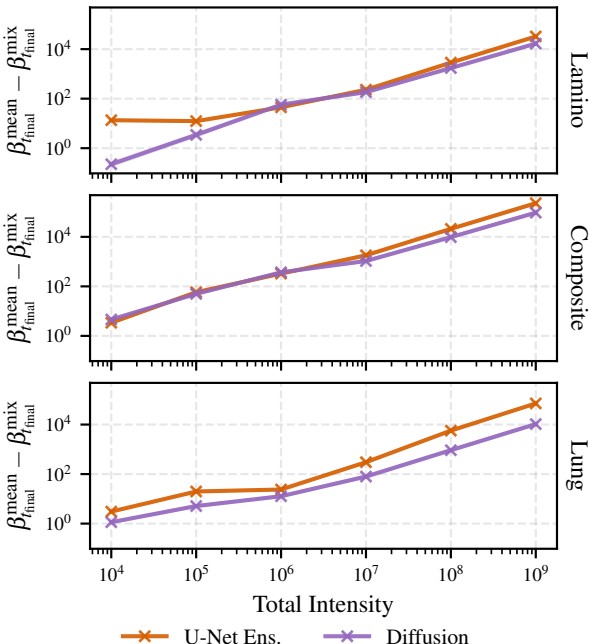

*Figure 6.* Difference between mean-based and mixture-based sequential negative log-likelihood at time step $t_{\text{final}} = 190$: $\beta_{t_{\text{final}}}^{\text{mean}} - \beta_{t_{\text{final}}}^{\text{mix}}$. Positive values indicate that the mixture distribution yields a lower sequential negative log-likelihood (better performance). Shaded regions indicate mean $\pm$ SEM (100 test set images, 10 seeds).

the final generated samples are both diverse and strictly statistically consistent with the observed data.

## D. Additional Results for the Sparse Setting

### D.1. Mixing and Mean-Aggregation

In this section, we analyze the theoretical and empirical differences between the *mixture* approach and the *mean-aggregation* method for constructing the sequential confidence bounds.

Recall that the confidence threshold $\beta_t$ is determined by the sequential negative log-likelihood of the prediction sequence. We compare two choices for the mixing distribution $\mu_{s-1}$ derived from a set of $K$ ensemble predictions or diffusion samples $\{\tilde{\mathbf{x}}_{s-1}^{(k)}\}_{k=1}^{K}$ or any $s = 1, \ldots, t$:

1. **Mean-Aggregation:** We collapse the ensemble into a single point estimate $\bar{\mathbf{x}}_{s-1} = \frac{1}{K} \sum_{k=1}^{K} \tilde{\mathbf{x}}_{s-1}^{(k)}$. The mixing distribution is a Dirac mass $\mu_{s-1}^{\text{mean}} = \delta_{\bar{\mathbf{x}}_{s-1}}$. The contribution to the confidence coefficient is:

$$\ell_s^{\text{mean}} = -\log p_{\bar{\mathbf{x}}_{s-1}}(\mathbf{y}_s \mid \alpha_s).$$

2. **Mixture:** We retain the full diversity of the samples using a uniform mixture $\mu_{s-1}^{\text{mix}} = \frac{1}{K} \sum_{k=1}^{K} \delta_{\tilde{\mathbf{x}}_{s-1}^{(k)}}$. The

contribution to the confidence coefficient is:

$$\ell_s^{\text{mix}} = -\log \left( \frac{1}{K} \sum_{k=1}^{K} p_{\tilde{\mathbf{x}}_{s-1}^{(k)}}(\mathbf{y}_s \mid \alpha_s) \right).$$

The difference in performance visualized in Figure 6 can be primarily attributed to the geometry of the image manifold and the non-linearity of the likelihood function.

**The Blurring Penalty.** In high-dimensional image spaces, the set of plausible reconstructions lies on a complex, often non-convex manifold. A well-calibrated ensemble or diffusion model produces samples $\{\tilde{\mathbf{x}}^{(k)}\}_{k=1}^{K}$ that lie on this manifold (i.e., they contain realistic high-frequency textures and sharp edges). However, the Euclidean mean $\bar{\mathbf{x}}$ of these diverse samples often falls off the manifold. In the context of CT, this manifests as *blurring*: sharp edges that vary slightly in position across ensemble members become smoothed out in the mean image.

When projected into the measurement space via the Radon transform, these smoothed edges fail to match the high-frequency transitions present in the observed sinogram $\mathbf{y}_t$. Because the Poisson likelihood is highly sensitive to residuals, especially in high-dose regimes where the noise floor is low, the mean image $\bar{\mathbf{x}}$ incurs a significant likelihood penalty.

**The Robustness of Mixtures.** In contrast, the mixture approach $\ell_s^{\text{mix}}$ utilizes a *log-sum-exp* structure. This acts as a soft-maximum operator over the likelihoods of the individual members. Mathematically, if even a single sample $\tilde{\mathbf{x}}^{\text{max}}$ in the mixture accurately models the local structure corresponding to the current projection angle $\alpha_t$, the term $p_{\tilde{\mathbf{x}}^{\text{max}}}(\mathbf{y}_s \mid \alpha_s)$ will be large. This dominant term effectively caps the negative log-likelihood penalty:

$$\ell_s^{\text{mix}} = -\log \left( \sum_{k=1}^{K} \exp \left( \log p_{\tilde{\mathbf{x}}_{s-1}^{(k)}}(\mathbf{y}_s \mid \alpha_s) \right) \right) + \log K$$
$$\approx -\log p_{\tilde{\mathbf{x}}_{s-1}^{\text{max}}}(\mathbf{y}_s \mid \alpha_s) + \log K$$

Consequently, the mixture approach hedges against individual model errors. It allows different ensemble members to explain different parts of the data sequence, providing a significantly tighter bound $\beta_t$. This explains the empirical gap observed in Figure 6, which widens at higher intensities ($10^9$) where the likelihood function becomes sharper and strictly penalizes the smoothing artifacts inherent in the mean-aggregation method.

### D.2. Empirical Crossover Rate Analysis

In this section, we analyze the relationship between the user-specified error tolerance $\delta$ and the empirical Type-I

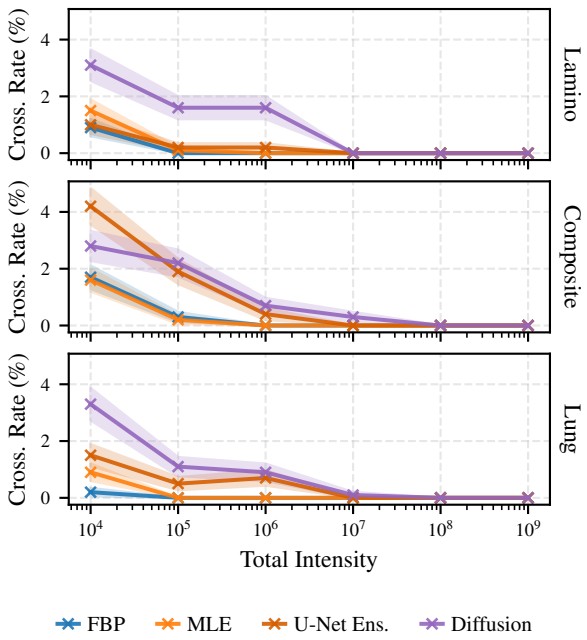

*Figure 7.* Crossover rate vs. total intensity for error level $\delta = 0.05$. Shaded regions indicate $\pm$ SEM (100 test set images, 10 seeds).

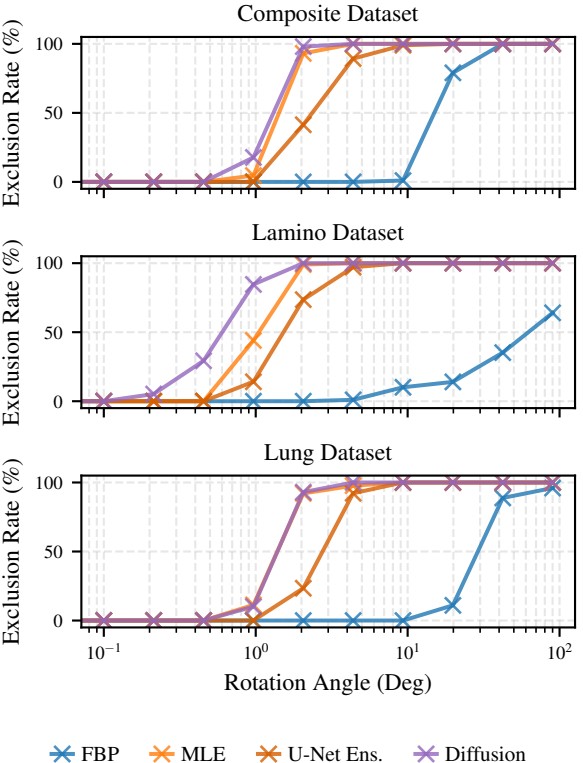

*Figure 8.* Exclusion rates vs. rotation angle for the ground truth image at total intensity $10^9$ and time step $t_{\text{final}} = 190$. Shaded regions indicate mean $\pm$ SEM (100 test set images, 10 seeds).

error rate. We define the *crossover rate* as the fraction of test sequences where the ground truth image $\mathbf{x}^*$ was excluded from the confidence set $C_t$ at least once during the acquisition horizon $t = 1, \ldots, 190$.

Figure 9, Figure 10, and Figure 11 visualize this relationship for the Diffusion, U-Net Ensemble, and MLE predictors, respectively. Across all methods, datasets, and intensities, the crossover curves remain strictly below the theoretical identity line (gray dashed). This confirms that our sequential likelihood mixing framework empirically satisfies the theoretical coverage guarantee $\mathbb{P}(\exists t \in \mathbb{N} : \mathbf{x}^* \notin C_t) \leq \delta$.

We observe a distinct transition in behavior governed by the total signal intensity. In the low-intensity regime ($10^4$ photons, purple lines), the crossover rates scale almost linearly with $\delta$. This indicates that when the setting is noise-dominated (aleatoric uncertainty), the sequential bound is highly active and utilizes the available error budget extensively. Conversely, as intensity increases to $10^9$ (yellow lines), the curves flatten toward zero. In this high-dose regime, although the confidence sets become geometrically much tighter and more informative (as established in Section 4.5), the likelihood landscape becomes extremely sharp. Consequently, the probability of the ground truth escaping the set drops, resulting in a robust guarantee that rarely produces empirical violations despite the shrinking volume of the sets.

The choice of mixing distribution affects how the error rate

scales with $\delta$. The Diffusion model tracks the diagonal most closely at low intensities, suggesting it is particularly aggressive in utilizing the error budget when noise is present. The U-Net Ensemble and MLE exhibit flatter curves, with the U-Net showing steeper scaling on the Composite dataset compared to the baseline MLE. It is particularly notable that for the Diffusion model, the crossover rates approach the theoretical limit even within a relatively short horizon of 190 steps. This demonstrates that the derived confidence coefficients $\beta_t$ are not loose theoretical artifacts, but practical bounds that actively constrain the uncertainty set.

### D.3. Uncertainty Image Evaluation Metrics

We compare the proposed uncertainty visualization methods discussed in Section 4.6 and Appendix C by analyzing their per-pixel confidence intervals. The two properties we investigate are (1) their width and (2) at what rate they contain their corresponding ground truth pixel. Figure 5 visualizes them for total intensity range $10^4$ to $10^9$ and all three datasets.

As compared to the other visualization methods, Diffusion $C_t$-Boundary Sampling achieves high coverage while providing tight confidence intervals.

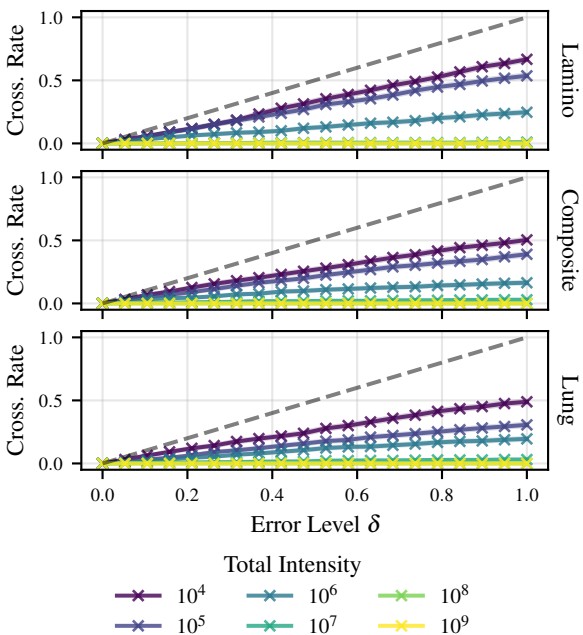

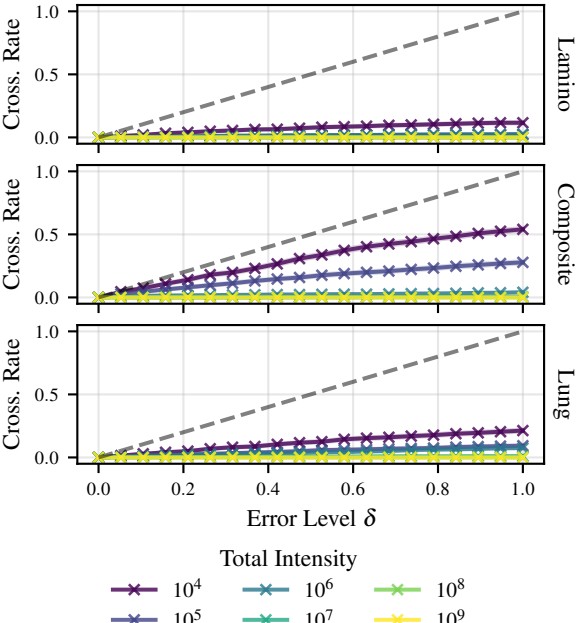

*Figure 9.* Crossover rate vs. level $\delta \in (0,1)$ using Diffusion model. The curves closest to the diagonal indicate the tightest valid bounds. Shaded regions indicate mean $\pm$ SEM (100 test set images, 10 seeds).

*Figure 10.* Crossover rate vs. level $\delta \in (0,1)$ using U-Net ensemble. The ensemble is more conservative than diffusion but outperforms MLE. Shaded regions indicate mean $\pm$ SEM (100 test set images, 10 seeds).

### D.4. Posterior-Sampling Baselines for Pixel-wise UQ

**Equivariant Bootstrap.** We follow Pereyra & Tachella (2024), using the FBP–U-Net of Section 4.2 (member 0) as the base estimator. For each $i \in \{1, \ldots, B\}$ with $B = 1000$: (1) draw a group element $g_i$ given by an in-plane rotation $\theta_i \sim \mathcal{N}(0, \sigma_{\text{rot}}^2)$, optionally composed with independent $\text{Bernoulli}(0.5)$ horizontal and vertical flips of the augmented signal; (2) simulate $\tilde{\mathbf{y}}_i$ from the Beer–Lambert–Poisson model in Section 2.1 applied to randomly transformed prediction $T_{g_i}\hat{\mathbf{x}}$; (3) reconstruct $\hat{\mathbf{x}}_i = \text{FBPUNet}(\tilde{\mathbf{y}}_i)$; and (4) invert the augmentation, $\tilde{\mathbf{x}}_i = T_{g_i}^{-1}\hat{\mathbf{x}}_i$. Pixel-wise percentile intervals are then computed over $\{\tilde{\mathbf{x}}_i\}_{i=1}^{B}$. The reference tomography setup of Pereyra & Tachella (2024) uses $\sigma_{\text{rot}} = 8°$ without flips. We calibrate $(\sigma_{\text{rot}}, \text{flip}) \in \{2, 4, 8, 16, 32, 64, 128\}° \times \{\text{false}, \text{true}\}$ per (dataset, total intensity) on the first 10 test images (image range $\{0, 1, \ldots, 9\}$, disjoint from the evaluation range $\{10, 11, \ldots, 109\}$) by minimising the absolute difference between the empirical and nominal coverage probability $1 - \delta$, averaged over images at $\delta = 0.05$, using $B = 100$ replicates during calibration.

**SK-ROCK.** We follow Pereyra et al. (2020) and draw approximate posterior samples given all $T = 200$ acquired projection angles $\alpha_1, \ldots, \alpha_T$. The target potential is

$$U(\mathbf{x}) = L(\mathbf{x}) + g^\tau(\mathbf{x}),$$

where $L(\mathbf{x}) = -\sum_{s=1}^{T} \log p_{\mathbf{x}}(\mathbf{y}_s \mid \alpha_s, I_0)$ is the Beer–Lambert Poisson NLL of Section 2.1 over all acquired angles, and $g^\tau : \mathcal{X} \to \mathbb{R}$ is the Moreau–Yosida envelope, with smoothing parameter $\tau > 0$, of $\mathbf{x} \mapsto w_{\text{TV}} \cdot \text{TV}(\mathbf{x})$ (isotropic total variation, weight $w_{\text{TV}} > 0$). The envelope is evaluated by 25 inner iterations of Chambolle's projection algorithm.

Following Durmus et al. (2018), we set $\tau = 1/M$, where $M \in \mathbb{R}_{>0}$ is an upper bound on the Lipschitz constant of $\nabla L$ on $\mathcal{X}$. Let $\mathbf{R} \in \{0,1\}^{(T \cdot r) \times r^2}$ denote the row-stacked Radon operator obtained by vertically concatenating $R_{\alpha_1}, \ldots, R_{\alpha_T}$. On $\mathcal{X} = [0,1]^{r \times r}$ we have $\exp(-(l/r)\mathbf{R}\mathbf{x}) \leq \mathbf{1}$ entry-wise, and a direct computation of the Hessian of $L$ yields $\|\nabla^2 L\|_2 \leq (l/r)^2 I_0 \|\mathbf{R}\|_2^2$. We set

$$M = 1.1 \cdot (l/r)^2 I_0 \|\mathbf{R}\|_2^2,$$

with a 10% safety margin to absorb the error of the $\|\mathbf{R}\|_2^2$ estimate, which we obtain by 40 steps of power iteration on $\mathbf{R}^\top \mathbf{R}$.

SK-ROCK is then run with $s = 10$ Chebyshev stages, damping $\eta = 0.05$, and step size $h = 0.95 \cdot \ell_s/(M + 1/\tau)$ where $\ell_s = (s - \frac{1}{2})^2(2 - \frac{4\eta}{3}) - \frac{3}{2}$. A single chain per image is initialised at the FBP reconstruction, projected onto $\mathcal{X}$ at every step, and run for 1000 burn-in iterations followed by 1000 retained samples. The TV weight $w_{\text{TV}}$ is selected per (dataset, total intensity) on the calibration

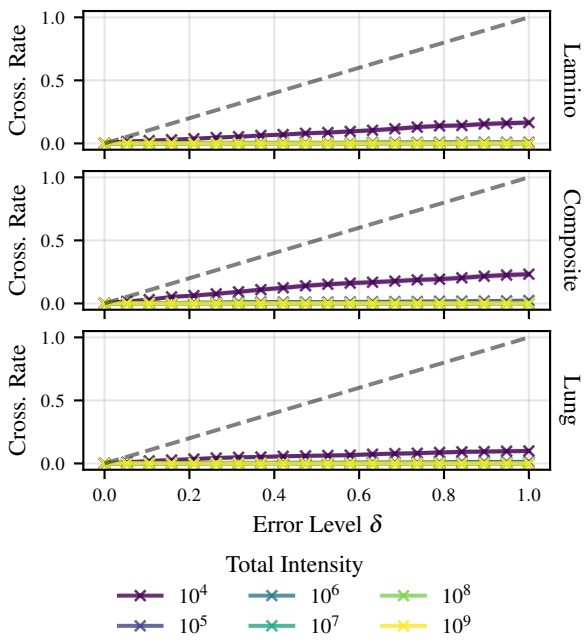

*Figure 11.* Crossover rate vs. level $\delta \in (0, 1)$ using MLE. The flat curves indicate highly conservative sets. Shaded regions indicate mean $\pm$ SEM (100 test set images, 10 seeds).

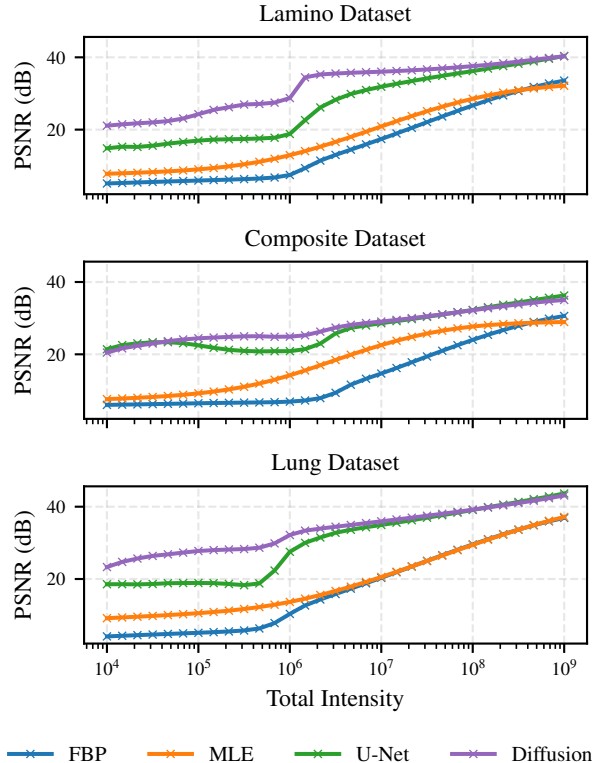

*Figure 12.* Reconstruction PSNR of the reconstruction vs. total intensity from dense observation data, sampled for 200 angles. Shaded regions (almost collapsed) indicate mean $\pm$ SEM (100 test set images, 10 seeds).

set (image range $\{0, 1, \ldots, 9\}$, disjoint from the evaluation range) over a logarithmic grid of 8 candidate values spanning two orders of magnitude, by minimising the expected calibration error averaged over the nominal coverage levels $\{0.1, 0.2, \ldots, 0.9, 0.95\}$.

## E. Sequential Dense-View Acquisition

In Section 2, we introduced a *sparse-view* acquisition protocol in which projection measurements are collected sequentially for a limited number of projection angles. In the sparse sampling regime, the inverse reconstruction problem is underdetermined. Although the coverage guarantee of Proposition 3.2 remains valid, meaningful confidence estimates cannot be obtained without imposing additional prior structure, e.g., by using a Diffusion prior.

To address this limitation and assess the impact of sparse measurement data on the performance, we now consider a *dense-view* acquisition protocol. We fix a finite set of $n$ measurement angles $\mathcal{A} = \{\alpha_1, \ldots, \alpha_n\}$, for example chosen as a uniform grid over $[0, 180)$. Let $(I_t)_{t \in \mathbb{N}}$ denote a sequence of incident intensities. At acquisition step $t$, we collect a full sinogram $\mathbf{y}_t \in \mathbb{N}_0^{n \times r}$, where

$$\mathbf{y}_{t,j} \sim p_{x^*}(\cdot \mid \alpha_j, n^{-1} I_t), \qquad j = 1, \ldots, n,$$

denotes the detector measurement corresponding to angle $\alpha_j$, and the total incident intensity over all angles at step $t$ is $I_t$. This formulation more closely reflects tomographic

acquisition with a rotating detector, which naturally gathers measurements jointly over a sequence of angles. The sequence $\{I_t\}$ represents successive exposures of the object, for instance modulated via exposure time or beam intensity.

Below, we parallel the evaluation presented in the main paper, using the same models and baselines described in Section 4.2, and the same setup and datasets as described in Section 4.1. The goal is to empirically validate sequential likelihood confidence sets in a setting where the inverse problem is well-posed and uncertainty arises *only* from observation noise.

For the angle space, we set 200 uniformly spaced angles. For the intensity sequence, we chose 30 steps on an exponentially spaced grid, starting at $I_1 = 10^4$ to $I_{30} = 10^9$ such that the cumulative total intensity at any step is log-uniformly distributed.

### E.1. Reconstruction Quality

As a sanity check, we compute PSNR reconstruction scores from the dense observation data, shown in Figure 12. As in the sparse setting, there is a clear ordering of methods as

expected, with the neural-network based approaches (Diffusion and U-Net) outperforming classical methods (FBP and MLE) by a large margin.

## E.2. Tightness of Confidence Sets

To evaluate the tightness of the confidence set, we plot the difference $\beta_t - L_t(\mathbf{x}^*)$ in the top row of Figure 13. Note that by the definition of the confidence set, it covers the ground truth image if $\beta_t - L_t(\mathbf{x}^*) \geq -\log\frac{1}{\delta}$. As in the sparse setting, using mixing distribution compared to mean point estimates leads to tighter confidence sets ($\beta_t^{(mix)} < \beta_t^{(mean)}$)) as shown in the second row of Figure 13. The difference is small ($< 1\%$) but consistent across datasets and the intensity range that we evaluated. We further rotated the ground truth image $\mathbf{x}^*_{rot}$ and tested the exclusion rate $\mathbf{x}^*_{rot} \notin C_t$ as a function of rotation angle in the bottom row of Figure 13. In other words, we are testing the power of the confidence set $C_t$ to rejects $\mathbf{x}^*_{rot}$ when the data is indeed generated from the true image $\mathbf{x}^*$. The highest exclusion rate is achieved by the Diffusion model, followed by the MLE and U-Net Ensemble.

Figure 14 shows the calibration curve of the confidence, i.e. the target failure probability $\delta$ over the empirical coverage rate. By design, the confidence set probability is conservative, i.e., Proposition 3.2 only provides an upper bound on the failure probability. Indeed, the size of the confidence set depends on the estimator's ability to predict the observation sequence. The tightest bounds are achieved by the Diffusion model, which coverage approaching the target coverage in the low-intensity regime where the observation noise is dominating. With higher intensities, the estimator's bias leads to increasingly more conservative bounds, with empirical coverage (for 100 test set images and 10 seeds each) approaches 1 around a total intensity of $10^6 - 10^7$. We remark that despite being conservative, the empirical coverage rate appears to be approximately a polynomial function of the nominal coverage. We leave empirical calibration of the confidence sets as an interesting direction for future work.

## E.3. Pixel-wise Confidence Intervals and Baselines

Finally, we compare pixel-wise confidence intervals computed using the worst-case optimization strategy (Appendix C.1) and Diffusion boundary sampling (Appendix C.2) to the same five baselines: Classical bootstrap sampling for FBP and U-Net reconstructions, U-Net ensembles, MCMC sampling using SK-Rock and Equivariant bootstrap as described in Appendix D.4. For all methods, confidence intervals are computed as 95% Gaussian confidence intervals based on the standard deviation of the samples or ensemble members respectively. Figure 15 shows mean coverage across pixels, mean width and Area Un-

der Sparsification Error (AUSE, Lind et al., 2024). Our results show that the U-Net ensemble achieve the smallest confidence intervals but with the lowest coverage. The bootstrapping variants achieve higher coverage, in particular, in the high-intensity (asymptotic) regime, however still below the nominal target. Diffusion $C_t$-Boundary Sampling and SK-ROCK achieve higher coverage with competitive confidence interval width, with the diffusion model achieving smaller confidence width in the low-intensity regime.

## E.4. Application: Hallucination Detection

As an application, we test the ability of the confidence set to detect hallucinations — plausible looking, but incorrect reconstructions. To have more fine-grained control over the hallucination rate, we trained an unconditional Diffusion model and varied the number of guidance steps. At each acquisition step, we generate samples $\tilde{\mathbf{x}}_t^{(1)}, \dots, \tilde{\mathbf{x}}_t^{(K)}$ from the Diffusion model, and check if $\tilde{\mathbf{x}}_t^{(k)} \in C_t$, where the confidence coefficient is computed with the conditional Diffusion model that achieves the tightest confidence coefficient (corresponding to higher power of the test). We note that by design, we control a type-I error: If $\tilde{\mathbf{x}}_t^{(k)} \notin C_t$, with probability at least $1 - \delta$ ($\delta = 0.05$, where the probability measure is over the randomness in the data), $\tilde{\mathbf{x}}_t^{(k)} \neq \mathbf{x}_t^*$. Proposition 3.2 does not give an explicit guarantee for the power of the test (the ability to reject a hallucination). Figure 16 shows the hallucination rate and PSNR computed over plausible samples ($\tilde{\mathbf{x}}_t^{(k)} \in C_t$) and hallucinated samples (defined by $\tilde{\mathbf{x}}_t^{(k)} \notin C_t$). The hallucination rate is the highest for unconditional diffusion with 5 gradient guidance steps per denoising step, followed by unconditional diffusion with 10 gradient steps per denoising step. The PSNR curves, conditioned on plausible and hallucinated samples respectively, show a clear gap in PSNR scores, demonstrating the ability of the confidence set to identify better reconstructions.

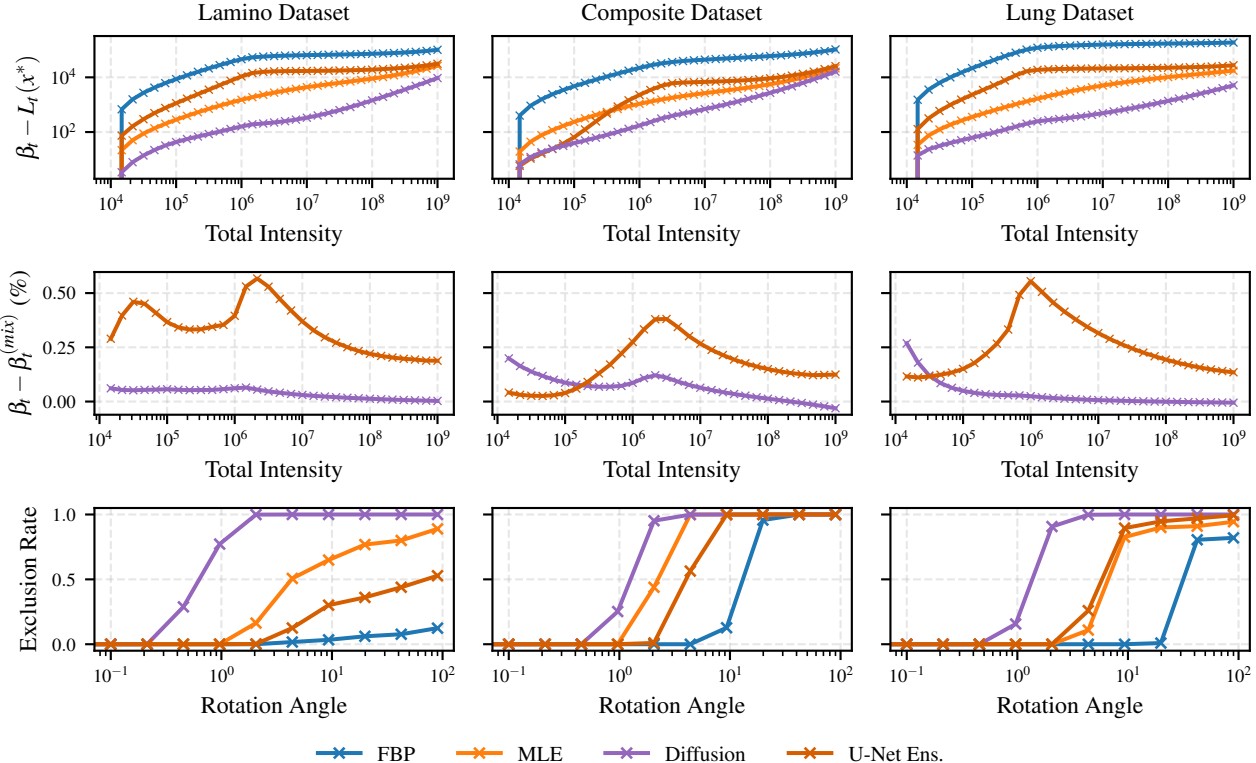

*Figure 13.* Tightness comparison of the sequential likelihood mixing confidence set for different reconstruction methods. The top row shows the confidence coefficient $\beta_t$ normalized by the negative log-likelihood of the true image. Lower values correspond to tighter confidence sets, demonstrating that the Diffusion-based confidence set outperforms classical methods. The middle row shows a comparison between the mixing based confidence coefficient, and a confidence coefficient based on the mean prediction. Larger values correspond to smaller mixing-based confidence coefficient. The results demonstrate that mixing yields a small but consistent advantage. The bottom row visualizes the exclusion rate of rotated version of the true image $\mathbf{x}^*_{rot} \notin C_t$ as function of the rotation angle. Higher exclusion rates correspond to larger power of the statistical test.

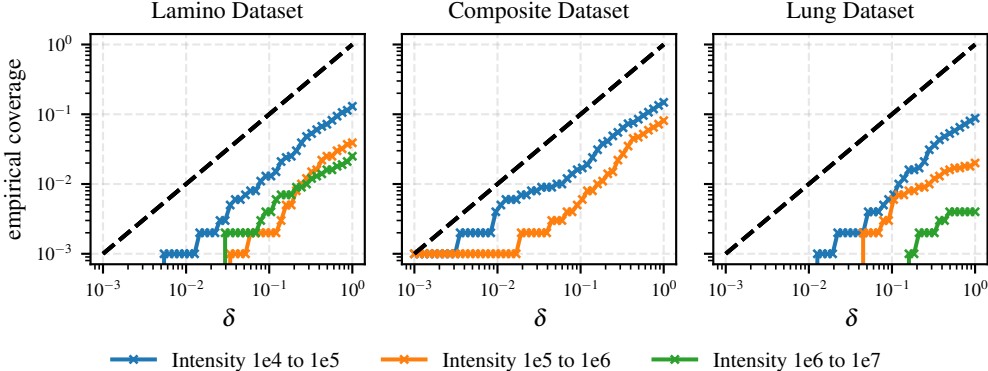

*Figure 14.* The calibration plots show empirical exclusion rates of the ground truth image $\mathbf{x}^*$ (Type-I error) over the target failure probability bound $\delta$. By design, the coverage rates are conservative. However, the construction achieves non-trivial coverage rates up to total intensities of $10^7$ where the statistical noise dominates. At higher intensities, the bias of the estimators dominates the magnitude of the confidence coefficient, an the empirical coverage rate (computed over 100 test set images and 10 seeds each) equals 1 (failure rate 0), hence are not included in the log-plots.

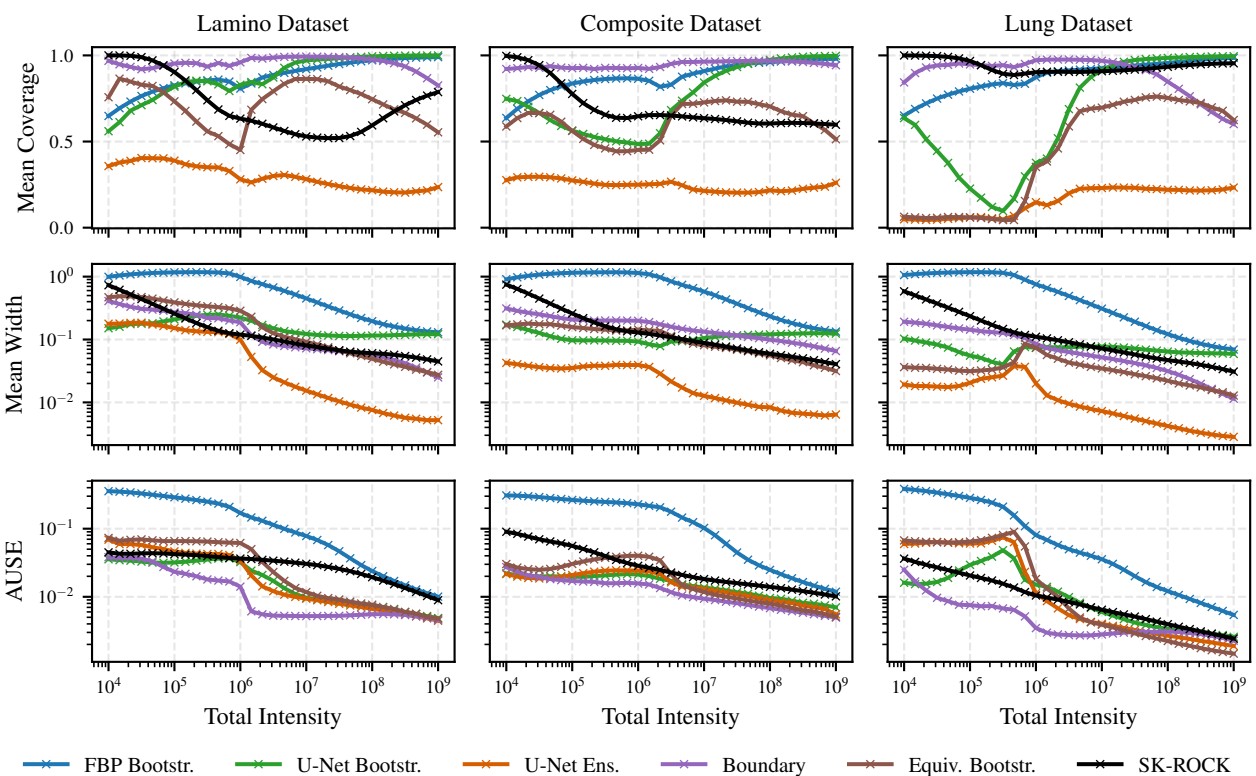

*Figure 15.* Mean coverage, mean width and Area Under Sparsification Error (AUSE) of pixel with confidence intervals computed for baselines (FBP and U-Net bootstrap with 100 samples, U-Net ensemble with 10 members), and derived from diffusion samples and $C_t$-Boundary guided Diffusion samples. Confidence intervals are based on Gaussian statistics of the samples at nominal level $\delta = 0.05$. U-Net ensemble display the lowest coverage, while being also the tightest. Both bootstrap estimators achieve good coverage at high intensities, where asymptotic statistics dominate. Diffusion $C_t$-Boundary Guided sampling improves coverage compared to standard Diffusion guidance, but also increase confidence width. Diffusion-based confidence intervals achieve the lowest AUSE compared to the baselines.

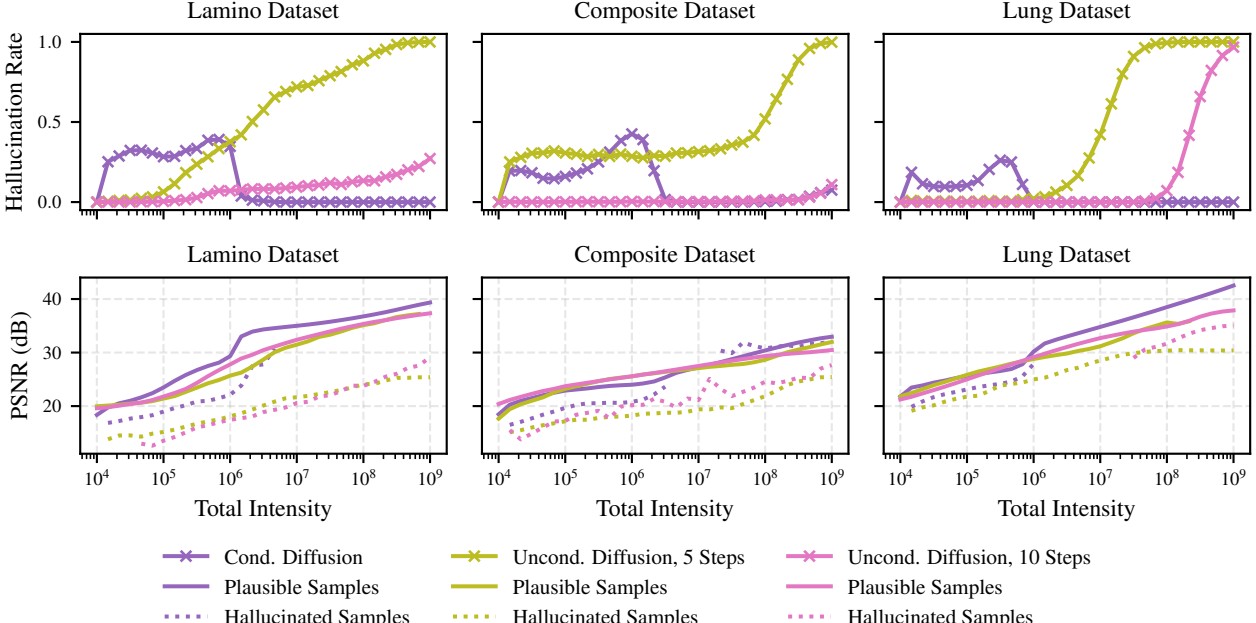

*Figure 16.* We use the confidence set defined for the conditional diffusion models to test if a generated sample satisfies $\tilde{\mathbf{x}}_t^{(k)} \in C_t$ at $\delta = 0.05$. Top row shows hallucination rates for an unconditional diffusion posterior sampling, with 5 and 10 guidance steps per denoising steps respectively, compared to the conditional diffusion model. Bottom row shows Peak Signal-to-Noise Ratio (PSNR) conditioned on plausible and hallucinated samples as defined by the confidence set. We note that sequential likelihood mixing by design controls only the type-I error (the probability to wrongly reject the ground-truth image), but not directly the power of the test (the probability to reject a incorrect reconstruction). In other words, if $\tilde{\mathbf{x}}_t^{(k)} \notin C_t$, indeed with high probability the test correctly identifies a hallucination, however, an image that satisfies $\tilde{\mathbf{x}}_t^{(k)} \in C_t$ may still display artifacts that are statistical plausible at the provided confidence level.

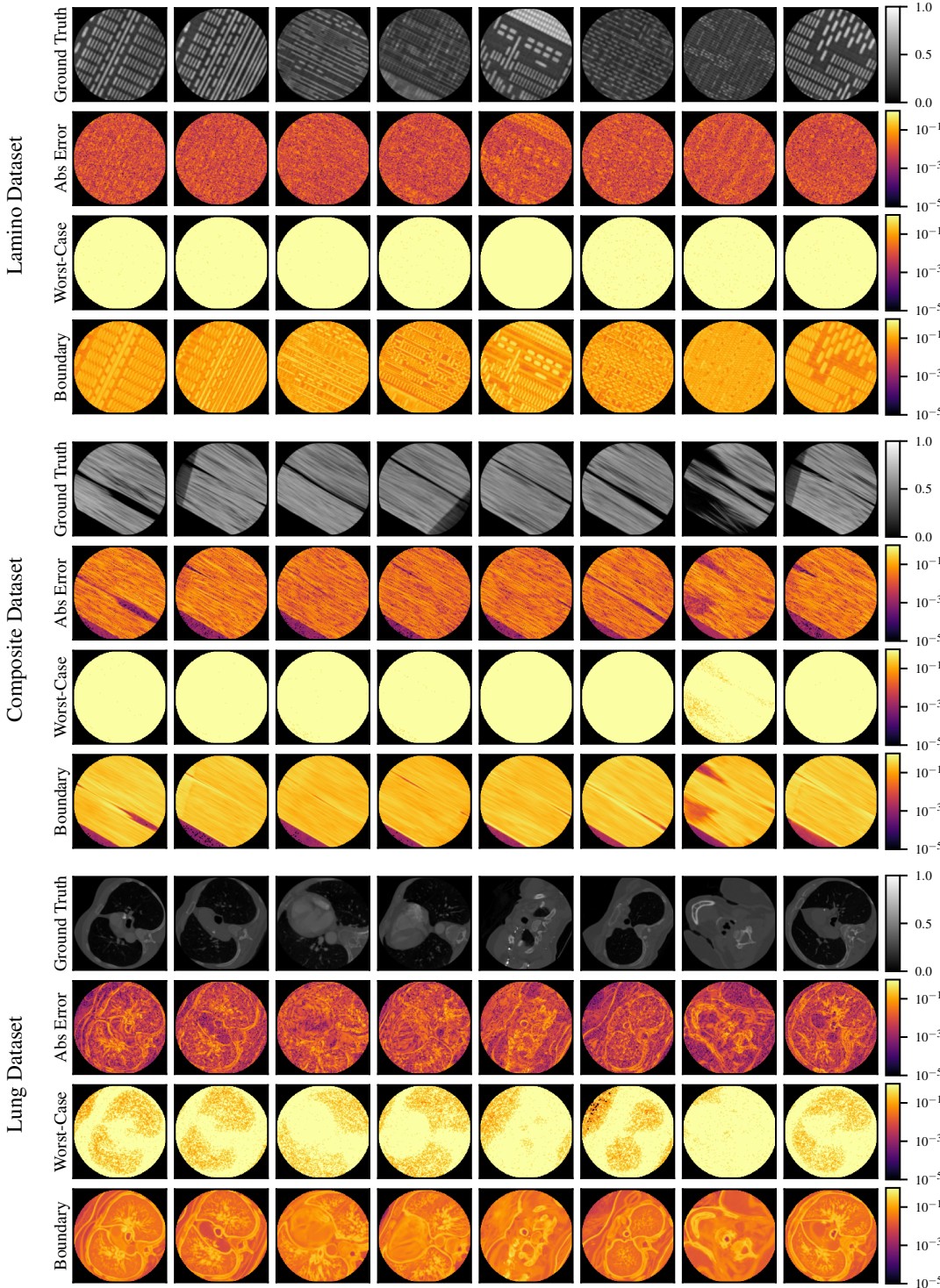

*Figure 17.* Comparison of worst-case uncertainty images and Diffusion $C_t$-Boundary Sampling uncertainty images with the absolute error for total intensity $10^6$. Uncertainty image pixel values correspond to half their corresponding confidence interval width to facilitate comparison with the visualized absolute errors. Absolute errors were computed between the means of conditional diffusion reconstructions and the ground truth image.

