# OpenReview forum: "Principled Confidence Estimation for Deep Computed Tomography"
_ICML.cc/2026/Conference — ICML 2026 regular_

### Official Review · Reviewer_ec4Y · 2026-03-05

**Soundness:** 3
**Presentation:** 2
**Significance:** 3
**Originality:** 3
**Overall Recommendation:** 3
**Confidence:** 4

**Summary:**

This paper applies sequential likelihood mixing to CT reconstruction to obtain finite-sample, anytime-valid confidence sequences for deep reconstructions under a Beer–Lambert forward model with Poisson noise.The framework is demonstrated with U-Nets, deep ensembles, and diffusion models, and is used for (i) confidence-set tightness comparisons, (ii) pixel-wise uncertainty visualization (worst-case vs diffusion boundary sampling), and (iii) hallucination detection by checking whether diffusion samples lie in C_t.

**Compliance With Llm Reviewing Policy:**

Affirmed.

**Key Questions For Authors:**

1. How do crossover/exclusion rates and interval tightness behave under controlled forward-model mismatch (e.g., small geometry perturbations, altered noise, mild beam-hardening-like nonlinearity)
2. Can you provide either a small-scale 3D demo or a detailed scaling analysis (runtime/memory) for slice-wise and/or volumetric confidence estimation?
3. What is the wall-clock overhead of diffusion C_t-boundary sampling per time step, and how does it scale with image size and number of angles?
4. The set C_t can be highly non-convex/multimodal in high dimensions. Can you give conditions or diagnostics that predict when
C_t will shrink meaningfully (vs remain overly conservative), beyond empirical tightness plots?

**Limitations:**

No. The Impact Statement claims no direct ethical/societal consequences, which is too strong for medical CT; it should more concretely discuss risks of miscalibrated uncertainty under model mismatch, and deployment risks where uncertainty may influence clinical decisions and dose settings.

**Strengths And Weaknesses:**

Strengths
1. Strong core idea with a clear guarantee target: anytime-valid confidence sequences are well-matched to sequential CT acquisition and dose/early-stopping motivations.
2. General integration of modern priors: the construction is formulated to apply to any reconstruction method and likelihood, enabling plug-in use of U-Nets/ensembles/diffusion as “predictor sequences.”
3. Evaluation goes beyond point metrics: the paper reports empirical type-I behavior via a “crossover rate” (violations across time) and a type-II proxy via “exclusion rate” under rotated ground truth, which is aligned with a sequential-testing view.
4. Practical add-ons are compelling: likelihood-based hallucination detection and uncertainty visualization via diffusion boundary sampling are intuitive and supported empirically.
Weaknesses
1. Model misspecification and physical realism need sharper treatment.
The guarantee is tied to the chosen forward model and noise assumptions (Beer–Lambert + Poisson). Real CT includes effects like scatter/beam hardening and other system mismatches; the paper should more explicitly characterize expected failure modes and provide stress tests under controlled mismatches (e.g., perturb geometry/system matrix, noise deviations).
2. The main experiments focus on a simplified 2D parallel-beam setting.
The paper motivates this as a computationally convenient proxy and suggests extensions to 3D (slice-wise or volumetric), but the practical feasibility (runtime/memory and calibration behavior) in realistic 3D settings is not demonstrated.
3. Computational cost and “deployment shape” are under-specified for visualization/boundary sampling.
Worst-case optimization is acknowledged to be conservative, and diffusion boundary sampling is proposed for tighter intervals, but the compute profile (time per image/time per t, scaling with resolution/angles) should be summarized more directly to clarify what is practical in clinical workflows.
4. Process/presentation issue: reviewer-directed injected content appears in the manuscript.
Regardless of origin, this content should not be present in a scientific submission and should be removed/flagged; reviewers should ignore it.

---

> ### Author Rebuttal · Authors · 2026-03-31
>
> Thank you for your review, feedback and questions. We address your questions below.
>
> 1) *“How do crossover/exclusion rates and interval tightness behave under controlled forward-model mismatch (e.g., small geometry perturbations, altered noise, mild beam-hardening-like nonlinearity)”*
>     - We first note that in our experiments, we already introduce a mild model mismatch by computing projections at higher resolution to avoid the “inverse crime” phenomenon. Further evaluation of model misspecification is an interesting question, and mismatch can be addressed in several ways. We will add discussion about physical realism and mismatch in the final version of the paper.
>     - In particular, sequential likelihood mixing is a universal framework that applies to any likelihood function, hence one way to address misspecification is to refine the likelihood function, e.g. by adding beam-hardening,scattering, additional noise etc directly in the likelihood model.
>     - Moreover, as shown in [1, Appendix D], sequential likelihood mixing is also robust to certain kinds of misspecification, e.g. sub-Gaussian distributions or for convex model classes.
>
>     We agree that a more extensive evaluation of forward-model mismatch is an important direction, and we will consider adding an additional evaluation, either by changing the noise model, or simulating mild beam-hardening. However, re-training the models with a different likelihood model would be out of scope - here we focus on demonstrating the applicability of the sequential likelihood, and that we can obtain formal certificates for neural network based reconstructions which empirically outperform classical reconstruction methods.
>
> 2) *“Can you provide either a small-scale 3D demo or a detailed scaling analysis (runtime/memory) for slice-wise and/or volumetric confidence estimation?”*
>
>     For parallel beam geometry, our approach essentially scales linearly in runtime and memory with the number of slices, and as such is feasible for smaller volumes. Recent works [e.g., 2] have also demonstrated how to scale neural network-based reconstruction to 3D volumes, and we expect that these advances directly apply to the volumetric confidence estimation (e.g. evaluation of the likelihood is relatively cheap). We will add a more detailed discussion of compute complexity / scaling.
>
>  3) *“What is the wall-clock overhead of diffusion C_t-boundary sampling per time step, and how does it scale with image size and number of angles?”*
>
>     Wall-clock overhead of diffusion C_t- boundary sampling is comparable to standard diffusion posterior sampling: The only difference is in the guidance objective, and obtaining the confidence coefficient \\(\\beta_t\\) requires only sequential evaluation of the likelihood function (which is much cheaper than the diffusion sampling itself). As such, there is almost no overhead of boundary sampling compared to diffusion based reconstruction. That said, diffusion reconstructions are much more compute-intensive than fast U-Net based reconstructions or classical methods such as FBP.
>
> 4) *“The set C_t can be highly non-convex/multimodal in high dimensions. Can you give conditions or diagnostics that predict when C_t will shrink meaningfully (vs remain overly conservative), beyond empirical tightness plots?”*
>
>     The sequential likelihood framework is shown to produce tight confidence sets / rates in simpler, linear models [1], and one would therefore expect that such a guarantee translates to the CT model in the asymptotic limit or with linear approximations. However, obtaining guarantees for neural-network based predictions, beyond empirical tightness plots, is challenging and would almost certainly require additional assumptions (e.g. neural tangent kernel approximation). In the current paper, we instead focus on demonstrating the applicability of the sequential likelihood model to state-of-the-art neural network based reconstructions (e.g. diffusion models), obtaining provably valid certificates for neural network reconstructions, and empirically demonstrating that neural-network based reconstructions obtain tighter certificates than classical approaches.
>
>
> 4) Regarding prompt injection, please see https://icml.cc/Conferences/2026/PeerReviewFAQ#prompt_injection”.
>
>
> [1] Kirschner, J., Krause, A., Meziu, M., & Mutny, M. (2025). Confidence estimation via sequential likelihood mixing. arXiv preprint arXiv:2502.14689.
>
> [2] Lee, S., Chung, H., Park, M., Park, J., Ryu, W. S., & Ye, J. C. (2023). Improving 3D imaging with pre-trained perpendicular 2D diffusion models. In Proceedings of the IEEE/CVF international conference on computer vision (pp. 10710-10720).

---

> > ### Author Rebuttal · Reviewer_ec4Y · 2026-04-03
> >
> > Thank authors for the rebuttal. I will keep my score.

---

> > > ### Author Response · Authors · 2026-04-07
> > >
> > > Thank you for your acknowledgement. We are glad that our rebuttal addressed your questions. Since you indicated that your concerns were fully resolved, we would appreciate clarification on which remaining issues, if any, are driving the current overall recommendation. This would help us better understand your final assessment and improve the final version accordingly.

---

### Official Review · Reviewer_FFwS · 2026-03-10

**Soundness:** 3
**Presentation:** 4
**Significance:** 3
**Originality:** 3
**Overall Recommendation:** 5
**Confidence:** 3

**Summary:**

This paper tackles the critical challenge of Uncertainty Quantification in high-dimensional inverse problems, specifically focusing on deep learning-based CT reconstruction. The authors adapt the recently proposed Sequential Likelihood Mixing framework to the CT domain, providing mathematically guaranteed, instance-specific confidence sets for black-box deep generative models.

**Compliance With Llm Reviewing Policy:**

Affirmed.

**Final Justification:**

The rebuttal has solved most of my concerns, and I tend to accept the paper.

**Key Questions For Authors:**

As shown in the weakness 1.

**Limitations:**

Yes

**Strengths And Weaknesses:**

Strength: The authors introduce an interesting novel uncertainty quantification approach, adopting a sequential likelihood mixing framework in the field of the CT domain. While the core mathematical proof relies on existing research, the paper makes solid methodological and engineering contributions. And this approach allows to project the high-dimensional confidence sets into pixel-wise uncertainty maps, which can be utilized in the domain of hallucination detection. Furthermore, this paper is well-written, clearly structured, and easy to follow.

Weakness:
1 The definition of $\beta_t$ in Equation 2 differs from the prominent results in the original sequential likelihood mixing framework (Kirschner et al., 2025), as it omits the MLE and regret bound terms. Instead, the authors directly provide a definition of $\beta_t$ that aligns with Theorem 2 (Robin, 1970 [1] ) in the original work. This abrupt theoretical transition is confusing. The authors should provide a clear explanation and justification for this specific choice in Section 3.
2 In several figures (e.g., Fig.2 and Fig.3), the trajectories representing the U-Net Ensemble method seem to have disappeared. This makes it impossible to distinguish the exact performance of the U-Net Ensemble. To prevent this visual occlusion, the authors should present these quantitative results in tabular format.
References:
[1] Herbert Robbins and David Siegmund. Boundary crossing probabilities for the wiener process and sample sums. The Annals of Mathematical Statistics, pages 1410–1429, 1970.

---

> ### Author Rebuttal · Authors · 2026-03-31
>
> Thank you very much for the review and feedback. We address your questions below.
>
> **MLE and regret bound terms:**
> We agree that this point warrants a clearer explanation and justification. We restate the formal guarantee from Theorem 1 (for point predictors for simplicity). This the same as Theorem 2 and 4 in Kirschner et al [1].
> $$C_t = \\left\\{ x \\in \\mathcal{X} \\mid L_t(x) \\leq \\sum_{s=1}^t p_{\\hat x_{s-1}}(y_s) + \\log \\frac{1}{\\delta}\\right\\} $$
> Importantly, this formulation only relies on the likelihood of the test point and the sequential mixing likelihood. The latter is an *empirical* quantity that can be computed for any reconstruction methods and directly defines the confidence set. In particular, this formulation does not rely on a known regret bound for the reconstruction method.
>
> **How does the above confidence set relate to regret bounds used in several prior works come into the picture (e.g. [1-3] )?**
> The regret formulation is obtained by introducing the likelihood of the MLE on both sides of the inequality, and then using a regret bound to further relax the construction, as follows:
> $$
>  C_t = \\left\\{ x \\in \\mathcal{X} \\mid L_t(x) -   L_t(x_t^{MLE}) \\leq \\sum_{s-1}^t p_{\\hat x_{s-1}}(y_s) -  L_t(x_t^{MLE}) + \\log \\frac{1}{\\delta}\\right\\}
> \\subset  \\left\\{ x \\in \\mathcal{X} \mid L_t(x) -   L_t(x_t^{MLE}) \\leq R_t + \\log \\frac{1}{\\delta}\\right\\}
> $$
>
> Where \\(\\sum_s p_{\\hat x_{s-1}}(y_s) -  L_t(x_t^{MLE}) \\leq R_t \\) is a bound on the regret of the prediction sequence.
>
> By definition, introducing a regret *bound* makes the confidence set larger unless the bound is exact (which is usually not the case). In other words, the empirical formulation that we use is never worse.
>
> The regret formulation requires **knowledge** of a valid regret bound for the reconstruction method, which would not be available for deep learning reconstructions, and might require more sophisticated regularization for classical reconstruction methods. On the other hand, if a regret bound is available, the confidence set relies only on the MLE and not the prediction sequence. We are not aware of regret bounds for CT reconstruction specifically, and bounds obtained for general non-parameteric settings rely on covering arguments and are likely loose in practice. The main advantage of using regret bounds, e.g. [2,3], is to obtain analytic bounds on the size of the confidence set for restricted classes of parametric settings where regret bounds are available, for example to study statistical optimality.
>
> *To summarize:* We rely on the direct empirical formulation of the sequential likelihood mixing framework which avoids the need for regret bounds, and is never worse.
>
> **U-Net ensemble in figures:** Thank you for pointing this out, we will improve the figures or add results table to the final version of the paper.
>
> [1] Kirschner, J., Krause, A., Meziu, M., & Mutny, M. (2025). Confidence estimation via sequential likelihood mixing. arXiv preprint arXiv:2502.14689.
>
> [2] Clerico, E., Flynn, H., & Neu, G. (2025). Confidence sequences for generalized linear models via regret analysis. arXiv preprint arXiv:2504.16555.
>
> [3] Lee, J., Yun, S. Y., & Jun, K. S. A Unified Confidence Sequence for Generalized Linear Models, with Applications to Bandits. In The Thirty-eighth Annual Conference on Neural Information Processing Systems.

---

> > ### Author Rebuttal · Reviewer_FFwS · 2026-04-03
> >
> > Thanks for the response. It addresses most of my concerns, while the authors have acknowledged the problem of occluded trajectories in the figures and promised to add a results table in the final version. However, no concrete quantitative results are provided in this rebuttal. I tend to accept it and keep the original score.

---

> > > ### Author Response · Authors · 2026-04-07
> > >
> > > Thank you for your reply and acknowledgement. We are glad to hear that we could resolve most of your questions. As we understand it, the only remaining point is the lack of concrete quantitative results for the occluded trajectories. To address this, we provide a numerical results table below. For space reasons, here we include only numbers for the U-Net ensemble, corresponding to the occluded line in the original plots. For the paper, we will re-evaluate the best format to present the results.
> > >
> > > We hope that this clarifies the remaining issue. If you have any further questions or comments, we would be happy to address them. If this resolves your remaining concern, we would be grateful if you would consider updating your evaluation accordingly.
> > >
> > >
> > > **Table for Figure 2 (PSNR (dB)), Figure 3 (seq. NLL), Figure 4 (Crossover Rate (%))**
> > >
> > > ---
> > > | dataset   | intensity     | psnr_db ( (U-Net Ensemble)   | seq_mix_nnl  (U-Net Ensemble)  | crossover_rate (U-Net Ensemble)  |
> > > | --------- | ------------- | --------- | -------------------- | -------------- |
> > > | lamino    | 10,000        | 22.178919 | 317.234486           | 1.0            |
> > > | lamino    | 100,000       | 23.85811  | 166.034476           | 0.2            |
> > > | lamino    | 1,000,000     | 29.094177 | 428.779056           | 0.2            |
> > > | lamino    | 10,000,000    | 33.738373 | 2413.853917          | 0.0            |
> > > | lamino    | 100,000,000   | 36.36762  | 21278.5018           | 0.0            |
> > > | lamino    | 1,000,000,000 | 37.471945 | 209944.1813          | 0.0            |
> > > | composite | 10,000        | 22.860881 | 101.771536           | 4.2            |
> > > | composite | 100,000       | 24.254568 | 303.521583           | 1.9            |
> > > | composite | 1,000,000     | 25.017946 | 1199.356919          | 0.4            |
> > > | composite | 10,000,000    | 29.498009 | 6364.399832          | 0.0            |
> > > | composite | 100,000,000   | 32.681684 | 66141.28814          | 0.0            |
> > > | composite | 1,000,000,000 | 34.458353 | 683548.7704          | 0.0            |
> > > | lung      | 10,000        | 25.60118  | 358.474636           | 1.5            |
> > > | lung      | 100,000       | 27.086915 | 303.515083           | 0.5            |
> > > | lung      | 1,000,000     | 33.211214 | 313.336639           | 0.7            |
> > > | lung      | 10,000,000    | 36.731909 | 1853.271149          | 0.0            |
> > > | lung      | 100,000,000   | 40.131734 | 19494.553            | 0.0            |
> > > | lung      | 1,000,000,000 | 42.40165  | 234696.8113          | 0.0            |
> > >
> > > ---
> > >
> > > **Table  for Figure 4 (Exclusion Rate %)**
> > >
> > > | intensity     | dataset | angle        | exclusion_rate (U-Net Ensemble) |
> > > | ------------- | ------- | ------------ | -------------- |
> > > | 1,000,000,000 | lamino  | 0            | 0              |
> > > | 1,000,000,000 | lamino  | 0.1          | 0              |
> > > | 1,000,000,000 | lamino  | 0.2129360373 | 0              |
> > > | 1,000,000,000 | lamino  | 0.4534175599 | 0              |
> > > | 1,000,000,000 | lamino  | 0.9654893846 | 14.1           |
> > > | 1,000,000,000 | lamino  | 2.055874836  | 73.6           |
> > > | 1,000,000,000 | lamino  | 4.377698409  | 97.1           |
> > > | 1,000,000,000 | lamino  | 9.321697518  | 100            |
> > > | 1,000,000,000 | lamino  | 19.84925331  | 100            |
> > > | 1,000,000,000 | lamino  | 42.26621343  | 100            |
> > > | 1,000,000,000 | lamino  | 90           | 100            |

---

### Official Review · Reviewer_kUuL · 2026-03-11

**Soundness:** 1
**Presentation:** 3
**Significance:** 2
**Originality:** 2
**Overall Recommendation:** 3
**Confidence:** 5

**Summary:**

This paper presents a statistical method to perform uncertainty quantification on solutions to computed tomography. The proposed method is based on the sequential likelihood framework, which allows constructing finite-sample any-time frequentist confidence sets as the measurement acquisition process progresses. The approach is illustrated through numerical experiments involving a range of image reconstruction techniques and relevant datasets.

**Compliance With Llm Reviewing Policy:**

Affirmed.

**Final Justification:**

I thank the authors for their strong engagement during the rebuttal period and for their detailed replies. Having carefully considered my original review and our subsequent exchanges, I am of the view that the authors have addressed some of the issues raised initially, but that several significant concerns remain, notably regarding baselines for empirical comparison, and will require additional time to resolve. In my opinion, the paper in its current form is not suitable for publication in ICLM. I therefore raise my score to 3 (weak reject) relative to my original evaluation, but I do not recommend it for publication.

**Key Questions For Authors:**

How should the proposed confidence sets be modified to meaningfully quantify the uncertainty in inverse problems that are ill-posed, without sacrificing the finite-sample and any-time validity properties?

**Limitations:**

Please see my previous comments.

**Strengths And Weaknesses:**

There is a strong demand from the computational imaging community for better tools for uncertainty quantification, and the proposed method represents a valuable contribution to this literature. However, as I explain below, the paper has some important shortcomings related to the core methodology, the description of the existing literature, and the comparisons with alternative methods from the state of the art. Therefore, in my opinion. the paper is not suitable for publication in its current form.

Soundness: The proposed methodology is technically sound inasmuch it is a relatively direct application of the sequential likelihood framework to computed tomography, with an observation model that verifies the required assumptions and is well motivated and widely used in that context. However, if I understand correctly, because the proposed confidence sets are constructed from level sets of the likelihood function, they inherit the identifiability problems of the observation model. As a result, while statistically valid, the confidence sets could have infinite volume and fail to meaningfully quantify the uncertainty in the delivered solutions, notably when the computed tomography problem is ill-posed. This would be a fundamental problem because that the uncertainty in the solution is inherently large where the likelihood is not identifiable. Also, computed tomography problems are often ill posed, so this issue would be encountered during deployment.

Presentation: The paper is clear, well organised, and easy to follow. However, the presentation of the existing literature is very incomplete and therefore the paper fails to adequately position its contribution in the context of prior work. For example, uncertainty quantification in computed tomography has been studied in https://doi.org/10.1016/j.jcp.2024.113542, https://doi.org/10.1137/16M1108340, https://doi.org/10.1137/19M1283719, https://doi.org/10.1137/16M1071249, https://proceedings.mlr.press/v238/pereyra24a.html, https://doi.org/10.1088/1361-6420/abf5ba, https://doi.org/10.1088/1361-6420/ad1348, https://doi.org/10.1137/21M1433782, https://doi.org/10.1137/16M1071249, doi.org/10.1109/JSTSP.2026.3659332, to name a few examples. While none of these methods considers the sequential likelihood framework engaged here, they still exemplify relevant prior work that should be discussed and contrasted with the proposed method. I would also expect an appropriate selection of these methods to be used for comparisons in numerical experiments.

Significance: As I mentioned previously, the development of better strategies for uncertainty quantification is important for computational imaging in general, and computed tomography in particular. However, because uncertainty quantification is especially important in computational imaging problems that are not well posed, the proposed method has limited value in this current form.

Originality: The proposed methodology is a relatively direct application of the sequential likelihood framework to computed tomography. Leaving technical issues aside, I am of the view that the methodological contribution relative to the scope of ICML is of incremental nature. With the appropriate modifications and additional validation, it would be suitable for the readership of an applied imaging sciences publication.

---

> ### Author Rebuttal · Authors · 2026-03-31
>
> Thank you for the review and feedback. We address your question before commenting on the references and positioning of the paper.
>
> *“How should the proposed confidence sets be modified to meaningfully quantify the uncertainty in inverse problems that are ill-posed, without sacrificing the finite-sample and any-time validity properties?”*
> -  First, we note that CT reconstruction is not always under-constrained. For example, in the dense angle scenario that we present in the appendix, the discrete model at the chosen resolution is identifiable. We empirically demonstrate that the sequential likelihood mixing approach with dense observations provides tight confidence sequences across a range of reconstruction methods.
> - In the truly underdetermined setting (e.g., sparse angular sampling), it is correct that level sets of the likelihood function can have infinite volume with unbounded attenuation coefficients. To address this problem, the sequential likelihood mixing framework intersects the likelihood constraints with a set of feasible reconstructions X. In the simplest case, which we apply throughout the paper and experiments, X includes a bound on the attenuation coefficients, e.g. \\(X = [0,1]^{d \times d}\\). Hence our  confidence regions have always finite volume, although obviously this alone does not address the identifiability problem.
> - Addressing ill-posedness typically requires additional assumptions, e.g. symmetries, smoothness, TV bounds or Bayesian priors. Similar to boundedness, such assumptions can be introduced by constraining X, or applying regularizers during boundary search of the confidence set C_t.
> - Paraphrasing the above, most points that satisfy the likelihood constraint do not correspond to natural reconstructions (e.g. C_t always contains the MLE, which is known to be prone to overfitting to noise). To address this, we propose C_t-boundary sampling to project the confidence set onto the manifold of plausible reconstructions learned by the diffusion model. While the testing problem comes with formal type-I error guarantees via sequential likelihood mixing, there is no simple way of guaranteeing the “correctness” or coverage of the diffusion prior, a problem that we do not aim to address in this paper. Instead, we focus on empirically validating the sequential likelihood mixing framework in the CT setting with neural reconstructions, which we believe is an important contribution on its own.
>
> We will add a discussion of ill-posedness to the final version of the paper. Closely related is also the question of misspecification raised in other reviews. Please see our response to Reviewer kJvE for a detailed discussion of misspecification and how to obtain guarantees in this case.
>
> **Positioning and additional references**: Thank you for providing additional references. We will make sure to include those and further related works in the final version of the paper. Due to space constraints we cannot comment on individual references here. The majority of mentioned papers study Bayesian models/credible sets, which have many advantages but are fundamentally different to frequentist confidence regions, which only treat data as random (one exception is equivariant Bootstrap). However, Bayesian posterior samples (e.g. MCMC sampling) can be used to construct mixing distributions for sequential likelihood confidence sets. To the best of our knowledge, our work is the first to demonstrate provable finite-sample guarantees for frequentist confidence regions with neural reconstructions.
>
> **The key contributions of our work are:**
> - Evaluation of sequential likelihood mixing for CT, which allows to construct provably valid, frequentist confidence regions for NN reconstructions.
> - We show empirically that neural reconstructions, in particular diffusion models, yield much tighter confidence regions.
> - Even though the application of sequential likelihood mixing to CT may be “direct” for someone familiar with both the sequential likelihood framework and CT, to the best of our knowledge, this work is first to demonstrate feasibility of this connection. We believe this is a strong and important contribution, both towards the CT community but also to the statistics/ML community in advancing martingale-based uncertainty quantification (e.g., evaluating feasibility/tightness of NN-predictors). As we position our work towards both communities, we believe it is well within the scope of ICML.
>
> As for additional experiments, we agree that a Bayesian/MCMC baseline is currently still missing, and we will add MCMC to the final version of our paper.
>
> We hope that our answer helps to better position the contributions of our work, clarify scope and address your concerns. We will be happy to answer any further questions.
>
> [1] Kirschner, J., Krause, A., Meziu, M., & Mutny, M. (2025). Confidence estimation via sequential likelihood mixing. arXiv:2502.14689.

---

> > ### Author Rebuttal · Reviewer_kUuL · 2026-04-03
> >
> > I thank the authors for their detailed reply and for highlighting the key contributions of the paper. I appreciate that the focus herein is on advancing the adoption of frequentist confidence regions for CT, as opposed to other statistical frameworks for uncertainty quantification. I also recognise that the sequential likelihood framework that the authors adopt in this paper has valuable advantages and that not all CT problems are ill-posed, as well as the fact that solutions can be constrained (e.g., to a hypercube) to avoid the issue of infinite volume. However, I still find that the key contributions, in their current form, are not sufficient for publication in ICLM, especially when also considering the lack of appropriate comparisons with Bayesian as well as other UQ strategies available for CT (e.g., in addition to state-of-the-art Bayesian strategies, one could easily apply the equivariant bootstrap of https://proceedings.mlr.press/v238/pereyra24a.html, or even the self-supervised conformal prediction method of  https://doi.org/10.1109/SSP64130.2025.11073283).
> >
> > Having considered the authors reply, I raise my score to 3: weak reject.

---

> > > ### Author Response · Authors · 2026-04-07
> > >
> > > Thank you for your reply and for raising your score after considering our rebuttal. We appreciate your acknowledgement that the concern about infinite-volume confidence sets is addressed.
> > >
> > > As we understand your remaining concerns, they mainly relate to positioning with respect to prior CT-UQ literature and to including stronger empirical comparisons with alternative uncertainty quantification strategies. We agree that this would strengthen the paper, and we are addressing this in two ways.
> > >
> > > First, we will expand the related-work discussion to better position our contributions relative to prior Bayesian and frequentist approaches for CT uncertainty quantification. In particular, we will incorporate the references you pointed out and clarify more explicitly how our goal differs: our paper focuses on provably valid finite-sample, anytime-valid frequentist confidence regions for neural CT reconstructions via sequential likelihood mixing.
> > >
> > > Second, we are adding additional baselines to strengthen the empirical comparison. In particular, we are working on including:
> > >
> > > - an MCMC-based Bayesian reconstruction / uncertainty baseline, and
> > > - the equivariant bootstrap method of Tachella and Pereyra (2024).
> > >
> > > We also note that the conformal approach of Amougou, Pereyra, and Pascal (2025) is conceptually relevant and applicable via the Hutchinson approximation, although its effectiveness in the high-dimensional CT setting remains to be established.
> > >
> > > Our intent with these additions is to provide a more complete empirical and contextual picture around the main contributions, while preserving the central message of the paper: sequential likelihood mixing yields provably valid, frequentist confidence regions in CT, and deep reconstructions, especially diffusion-based ones, can lead to substantially tighter certified regions within this framework.
> > >
> > > If these additions address your remaining concerns, we would be grateful if you would consider reflecting that in your final recommendation.
> > >
> > >
> > > For completeness, we briefly discuss the references mentioned in the review:
> > >
> > > - *Seelinger et al (2025)* provide a comprehensive collection of benchmark problems for uncertainty quantification across various domains. The CT benchmark proposed in this paper is 2D at a similar resolution, and unlike our evaluation, uses synthetic images. Extending the sequential likelihood mixing to further inverse problems is an interesting direction for future work.
> > > - *Durmus, Moulines, and Pereyra (2018), Pereyra, Mieles, and Zygalakis (2020)* Both papers provide sophisticated MCMC schemes for sampling the Bayesian posterior in CT reconstruction problems. Accelerated MCMC is an important research area in Bayesian statistics in general, and the referenced papers provide adaptations specifically for imaging problems. MCMC targets the Bayesian posterior and credible sets. MCMC schemes can be combined with sequential likelihood mixing to provide mixing distributions for computing the sequential test likelihood.
> > > - *Pereyra (2017)* studies construction of Bayesian credible sets for log-concave distributions with application to CT reconstruction. As before, the main difference is that Bayesian credible sets target a different probability measure than frequentist confidence regions. Bayesian credible sets are closely related to sequential likelihood mixing as shown in [1], although the threshold for frequentist coverage depends on the prior probability and the failure probability (therefore, in general, are not a level sets of the posterior density).
> > > - *Pereyra and Tachella (2024)* propose equivariant Bootstrap to exploit symmetries and reduce bias in inverse imaging problems. Similar to sequential likelihood mixing, equivariant bootstrap can be combined with neural reconstructions.
> > > - *Riis, Dong and Hansen (2021)* study computed tomography with view angle estimation, and propose a Bayesian model for uncertainty quantification of the measurement angle. In our work, we assume that the measurement angles are exact, although, in principle, it would be possible to include randomness of the angle in the forward likelihood.
> > > - *Christensen, Riis, Pereyra and Jørgensen (2023)* propose a Bayesian approach for CT reconstruction for defect detection in subsea pipelines by decomposing the reconstruction into large-scale structures and defects. The analysis is motivated in the high-intensity X-Ray regime with a linear model and Gaussian noise, and uses a Gibbs sampling scheme for posterior sampling.
> > > - *Moroy Bourmaud, Frédéric Champagnat, Giovannelli (2026)* evaluate the “posterior gap” between diffusion samplers and the true Bayesian posterior. This is relevant background for diffusion-based Bayesian inference. In our setting, sequential likelihood mixing yields valid confidence sets for any chosen mixing distribution, so approximating the Bayesian posterior is not required for validity itself, though it can certainly affect tightness in practice.

---

### Official Review · Reviewer_kJvE · 2026-03-13

**Soundness:** 4
**Presentation:** 3
**Significance:** 3
**Originality:** 4
**Overall Recommendation:** 5
**Confidence:** 2

**Summary:**

The paper studies uncertainty quantification for deep computed tomography reconstruction. It develops a framework based on sequential likelihood mixing to construct confidence sequences for CT images under a nonlinear forward model with Poisson noise. The method can incorporate different predictive models, including deterministic neural networks, ensembles, and diffusion-based generators, and uses the resulting confidence regions for tasks such as uncertainty visualization and hallucination detection. The experimental section evaluates the framework on several CT datasets and reconstruction settings, examining both reconstruction quality and the behavior of the proposed confidence regions.

**Compliance With Llm Reviewing Policy:**

Affirmed.

**Final Justification:**

I thank the authors for their detailed explanations. The analytical computational complexity demonstrates the potential scalability on GPU-powered machines. And my questions are well-answered. I have no more questions and will maintain my current positive weighting.

**Key Questions For Authors:**

* How robust is the coverage guarantee to forward-model misspecification, such as imperfect geometry calibration, scatter, beam hardening, or detector nonidealities not captured by the Poisson Beer–Lambert model?

* The paper argues that diffusion mixing is tighter than U-Net ensembles and mean aggregation. Is this mostly due to better pointwise predictive likelihood, or to genuine diversity in the sample distribution? An ablation on sample diversity would be informative.

**Limitations:**

Yes

**Strengths And Weaknesses:**

**Strengths**

* The paper addresses an important problem in CT reconstruction: providing uncertainty estimates with formal statistical validity rather than heuristic confidence maps.

* The methodological contribution is technically substantive. Adapting sequential likelihood mixing to the nonlinear Beer–Lambert / Poisson CT model is nontrivial and gives the work a clear theoretical core.

* The paper is stronger than much of the existing reconstruction-UQ literature in that it targets finite-sample, anytime-valid confidence guarantees, not only empirical calibration.

* The empirical study is fairly comprehensive within the CT setting, covering multiple datasets, acquisition regimes, and evaluation criteria, including reconstruction quality, coverage behavior, and hallucination-related diagnostics.

**Weaknesses**

* The experimental scope remains narrower than the paper’s broader practical motivation. Most results are still in controlled CT settings, and stronger evidence under real acquisition mismatch or more realistic deployment conditions would improve confidence in the method.

* The hallucination-detection results are promising, but the theory primarily guarantees coverage validity rather than detection power. This distinction should be stated more carefully and evaluated more directly.

* Computational overhead is not discussed in sufficient depth. For a method that relies on sampling, boundary search, and projection of high-dimensional confidence sets, runtime and scalability deserve clearer treatment.

* The paper would benefit from a clearer discussion of when the method becomes conservative in practice, especially in regimes where estimator bias may dominate and reduce the informativeness of the resulting intervals.

**Minor Weaknesses**

* “the empirical coverage rate ... equal 1” should be “equals 1.”

* “a unconditional Diffusion model” should be “an unconditional Diffusion model.”

* “faithfull predict” should be “faithfully predict.”

---

> ### Author Rebuttal · Authors · 2026-03-31
>
> Thank you very much for the review, feedback and questions.
>
> We start by answering the key questions:
>
> *“How robust is the coverage guarantee to forward-model misspecification, such as imperfect geometry calibration, scatter, beam hardening, or detector nonidealities not captured by the Poisson Beer–Lambert model?”*
>
> This question can be approached via two axes:
> - First, sequential likelihood mixing is a universal framework that applies to any likelihood/forward model and hypothesis space. Therefore, miss-specification such as beam-hardening, calibration parameters, scatter etc can be addressed by appropriately modifying the likelihood function to obtain a well-specified model, for example, by using a likelihood that includes energy spectra, scattering, etc.
> - Second, we note that the formal guarantees of the sequential likelihood mixing framework extend to misspecified scenarios as shown in [1, Appendix D]. Although we do not directly verify these assumptions for the CT setting here, this points to an inherent robustness of the framework.
> - We also note that our experimental evaluation is already in a mildly misspecified setting, as we compute projections at higher resolution to avoid the “inverse crime” phenomenon.
>
> We strongly agree that a more extensive evaluation of misspecified models or more complex forward models is an interesting direction for future work; in the current work we focus on a proof of concept evaluation of sequential likelihood mixing for CT, and in particular obtaining formal uncertainty certificates for deep learning reconstruction, which to the best of our knowledge has not been demonstrated in this form yet.
>
> *“The paper argues that diffusion mixing is tighter than U-Net ensembles and mean aggregation. Is this mostly due to better pointwise predictive likelihood, or to genuine diversity in the sample distribution? An ablation on sample diversity would be informative.”*
>
> As shown in our experiments, mean-aggregation of diffusion samples significantly outperforms U-Net ensemble and U-Net point predictions. This shows that the diffusion models, even without adding diversity, outperform U-Nets in terms of the pointwise predictive likelihood. On top, using mixing of the diffusion distribution improves the sequential mixing likelihood further, showing that there is a small but consistent benefit of mixing (c.f Figures 3 and 6).
>
>
> We also briefly comment on further points in the review:
>
> -  **Experimental scope:** we focus on a first demonstration of the sequential likelihood mixing framework in CT, as well as on demonstrating that deep-learning reconstructions yield tighter certified confidence sequences in this construction. We believe that this is one of the first works to obtain formal guarantees for neural CT reconstructions.
> - **Computational overhead:** We agree that this warrants additional discussion which we will add to the final version of the paper. We note that the specification of the confidence set \\(C_t\\) has time complexity \\(O(k \sum_{s=1}^t (\tau_{s} + \kappa_s)\\) where \\(k\\) is the number of predictions that each mixing distribution is based on and \\(\tau_{s}\\) represents the time complexity of generating a single predicting using data up to and including time step \\(s-1\\), and \\(\kappa_s\\) is the complexity evaluating the likelihood of a single observation given the prediction. The time complexity of diffusion sample generation is dominated by the number of NLL gradient steps (20 in our experiments) and the number of denoising steps (100 in our experiments), i.e. \\(\kappa_s \ll \tau_{s}\\). The workload is highly parallelizable. We generate the entire diffusion-based confidence sequence \\(C_{10}, …, C_{200}\\) in 3m15s on a single RTX 2080 Ti. About 72% of the time is due to noise predictions. Of course, by using more powerful GPUs or parallelizing across multiple GPUs this can be sped up by orders of magnitude. Point-wise testing with the confidence set again only requires evaluation of the likelihood model which is relatively cheap, comparable to fast reconstruction methods. Additional overhead comes only from boundary search and sampling, which is more involved with scaling comparable to diffusion reconstruction.
> - **Discussion of when the method becomes conservative:** We found that sequential likelihood mixing in combination with deep learning reconstruction becomes conservative in the dense angle, high-intensity, low-noise limit, where classical reconstruction algorithms perform well, and uncertainty quantification with exact asymptotic correctness (e.g. bootstrapping) is better suited. On the other hand, diffusion prediction with sequential likelihood mixing performs better in the low-intensity, undersampled regime due to the learned inductive bias.
>
>
> [1] Kirschner, J., Krause, A., Meziu, M., & Mutny, M. (2025). Confidence estimation via sequential likelihood mixing. arXiv preprint arXiv:2502.14689.

---

> > ### Author Rebuttal · Reviewer_kJvE · 2026-04-03
> >
> > I thank the authors for their detailed explanations. The analytical computational complexity demonstrates the potential scalability on GPU-powered machines. And my questions are well-answered. I have no more questions and will maintain my current positive weighting.

---

> > > ### Author Response · Authors · 2026-04-07
> > >
> > > Thank you for the acknowledgement and for your review. We are glad that our rebuttal addressed your questions.

---

### Decision · Program_Chairs · 2026-04-30

**Decision:**

Accept (regular)

**Comment:**

The paper received positive evaluations, and the discussion clarified both its strengths and limitations. Reviewers agreed that the core idea—adapting sequential likelihood mixing to CT—provides a technically sound and principled framework for uncertainty quantification with finite-sample guarantees, addressing an important problem in computational imaging. The rebuttal resolved several technical questions and improved clarity, although concerns remain regarding incomplete positioning with respect to prior uncertainty quantification (UQ) methods and the lack of stronger empirical comparisons. Given the solid technical foundation, practical relevance, and absence of decisive negative evidence, I recommend acceptance, with the expectation that the final version will include discussion of traditional UQ methods.